EMBO
Molecular Medicine

# Comparative intravital imaging of human and rodent malaria sporozoites reveals the skin is not a species-specific barrier

Christine S Hopp[1,2,3,*,†] (ID), Sachie Kanatani[1,2,†], Nathan K Archer[4], Robert J Miller[4], Haiyun Liu[4],
Kevin K Chiou[5], Lloyd S Miller[4,6,‡] & Photini Sinnis[1,2,6,**] (ID)

## Abstract

Malaria infection starts with the injection of *Plasmodium* sporozoites into the host's skin. Sporozoites are motile and move in the skin to find and enter blood vessels to be carried to the liver. Here, we present the first characterization of *P. falciparum* sporozoites *in vivo*, analyzing their motility in mouse skin and human skin xenografts and comparing their motility to two rodent malaria species. These data suggest that in contrast to the liver and blood stages, the skin is not a species-specific barrier for *Plasmodium*. Indeed, *P. falciparum* sporozoites enter blood vessels in mouse skin at similar rates to the rodent malaria parasites. Furthermore, we demonstrate that antibodies targeting sporozoites significantly impact the motility of *P. falciparum* sporozoites in mouse skin. Though the sporozoite stage is a validated vaccine target, vaccine trials have been hampered by the lack of good animal models for human malaria parasites. Pre-clinical screening of next-generation vaccines would be significantly aided by the *in vivo* platform we describe here, expediting down-selection of candidates prior to human vaccine trials.

**Keywords** intravital; malaria; *Plasmodium*; skin; sporozoites
**Subject Categories** Methods & Resources; Microbiology, Virology & Host Pathogen Interaction
See also: **A Vaughan** (April 2021)

## Introduction

Malaria, the most deadly parasitic infection of humans, is caused by protozoan parasites of the genus, *Plasmodium*, affecting over 200 million people globally per year (World Health Organization, 2018). The majority of deaths due to malaria are caused by *Plasmodium falciparum*, which is endemic in sub-Saharan Africa. The symptoms of malaria are caused by cyclic multiplication of the parasite in host erythrocytes, yet infection begins, when an infected mosquito inoculates *Plasmodium* sporozoites as it searches for blood (Sinnis & Zavala, 2012). Inoculated sporozoites actively migrate through the dermis to enter blood vessels (Hopp *et al*, 2015), where the circulation carries them to the liver. Sporozoites invade and multiply inside hepatocytes, from which thousands of liver merozoites are released to initiate blood stage infection.

Gliding motility is a striking feature of Apicomplexan parasites. Of the invasive stages of *Plasmodium*, sporozoites are the most impressive, moving at high speed (1–3 μm/s) for 1 h, frequently longer. Sporozoite motility is essential for their exit from the dermis and consequently for their infectivity (Vanderberg & Frevert, 2004; Amino *et al*, 2006; Hellmann *et al*, 2011; Ejigiri *et al*, 2012; Hopp *et al*, 2015). Intravital fluorescence microscopy of sporozoites moving through the dermis of laboratory mice is a powerful technique and has permitted qualitative and quantitative studies of the rodent malaria parasite *Plasmodium berghei*: its migration through the skin and its interaction with blood vessels (Vanderberg & Frevert, 2004; Amino *et al*, 2006; Hellmann *et al*, 2011; Hopp *et al*, 2015). These and other studies demonstrate that the dermal inoculation site is where the malaria parasite is extracellular for the longest period of time in the mammalian host, making this a time of vulnerability for the parasite.

Our previous work highlighted significant changes in *P. berghei* sporozoite motility over the first 2 h after inoculation (Hopp *et al*,

1 Department of Molecular Microbiology and Immunology, Johns Hopkins Bloomberg School of Public Health, Baltimore, MD, USA
2 Johns Hopkins Malaria Institute, Johns Hopkins Bloomberg School of Public Health, Baltimore, MD, USA
3 Laboratory of Immunogenetics, National Institute of Allergy and Infectious Diseases, National Institutes of Health, Rockville, MD, USA
4 Department of Dermatology, Johns Hopkins University School of Medicine, Baltimore, MD, USA
5 Department of Physics and Astronomy, University of Pennsylvania, Philadelphia, PA, USA
6 Department of Medicine, Division of Infectious Diseases, Johns Hopkins University School of Medicine, Baltimore, MD, USA
*Corresponding author. E-mail: christine.hopp@nih.gov
**Corresponding author. Tel: +1 410 502 6918; E-mail: psinnis1@jhu.edu
†These authors contributed equally to this work
‡Present address: Immunology, Janssen Research and Development, Spring House, PA, USA

2015). Upon entering the dermis, sporozoites move with high speed on relatively linear paths, likely to optimize their dispersal. After 20 min in the tissue, sporozoite trajectories become increasingly confined, a motility pattern that likely optimizes contact with blood vessels and blood vessel entry.

Due to emerging drug resistance of *Plasmodium* parasites and insecticide resistance in the mosquito vectors, a highly effective vaccine is widely viewed as a key step toward defeating malaria (Hopp & Sinnis, 2015; Long & Zavala, 2016). Sporozoite transmission is a significant bottleneck for the parasite, with 10–100 parasites being inoculated into the skin and only 20% of these successfully exiting the dermis (Medica & Sinnis, 2005; Yamauchi *et al*, 2007; Hopp & Sinnis, 2015), making this early life cycle stage an attractive target. Indeed, it has long been appreciated that vaccination with attenuated sporozoites confers protection, a finding that provided the foundation for the development of RTS,S, a subunit vaccine based on the sporozoite's major surface protein. In Phase III clinical trials RTS,S conferred partial protection against clinical disease and severe malaria (RTS,S Clinical Trials Partnership, 2014). Though falling short of community-established goals, this promising first step validated the sporozoite as a vaccine target, with follow-up studies demonstrating that protection was mediated by antibodies targeting the sporozoite's major surface protein, the circumsporozoite protein (CSP; Foquet *et al*, 2014). Previous studies demonstrated that antibodies targeting CSP can immobilize sporozoites *in vitro* and *in vivo* and can exert a large proportion of their protective efficacy in the dermal inoculation site (Vanderberg & Frevert, 2004; Foquet *et al*, 2014; Flores-Garcia *et al*, 2018; Aliprandini *et al*, 2018). Though gliding motility is a good target for interventions, it has only been studied in the rodent malaria parasite, and only *P. berghei* has been subject to quantitative intravital imaging (Vanderberg & Frevert, 2004; Amino *et al*, 2006; Hellmann *et al*, 2011; Hopp *et al*, 2015). Here, we present the first *in vivo* motility assessment of *P. falciparum* in mouse skin and grafted human skin in a humanized mouse model and compare the motility of human and rodent

malaria sporozoites *in vivo*. Using two rodent malaria species, *P. berghei* and *P. yoelii*, we also describe species-specific differences between the two rodent species. Moreover, we establish a protocol for high-throughput automated analysis of sporozoite motility *in vivo*. These studies establish an *in vivo* platform for the pre-clinical testing of vaccine candidates, monoclonal antibodies, and prophylactic drugs targeting the sporozoite stage of the malaria parasite.

## Results

### Imaging *Plasmodium yoelii* and *Plasmodium falciparum* in the rodent dermis

We first determined whether *P. falciparum* sporozoites are motile in mouse skin and set out to compare their motility to the rodent malaria parasites, *P. yoelii* and *P. berghei*. In order to image *P. yoelii* and *P. falciparum* sporozoites in the dermis, we made new transgenic parasite lines expressing a fluorophore under a strong sporozoite promoter such that sporozoites were sufficiently bright to be visualized by intravital microscopy. We generated a *P. yoelii* line expressing mCherry under control of the *P. berghei csp* promoter (PBANKA_0403200) (Appendix Fig S1) and a *P. falciparum* line expressing tdTomato under control of the *P. falciparum peg4* promoter (PF3D7_1016900) (McLean *et al*, 2019). Intravital microscopy of *P. yoelii* and P. falciparum sporozoites in the dermis of mice was performed as previously described for *P. berghei* (Hopp *et al*, 2015). Figure 1 shows maximum-intensity projections of sporozoite motility over 4 min in representative videos started 10 min after sporozoite inoculation (see Movies EV1–EV3 for the corresponding time-lapse video). As shown, *P. falciparum* sporozoites are motile in mouse skin. Dermal vasculature was visualized by labeling the pan-endothelial junction molecule CD31 by intravenous injection of fluorescently labeled rat anti-CD31 (Formaglio *et al*, 2014; Hopp *et al*, 2015).

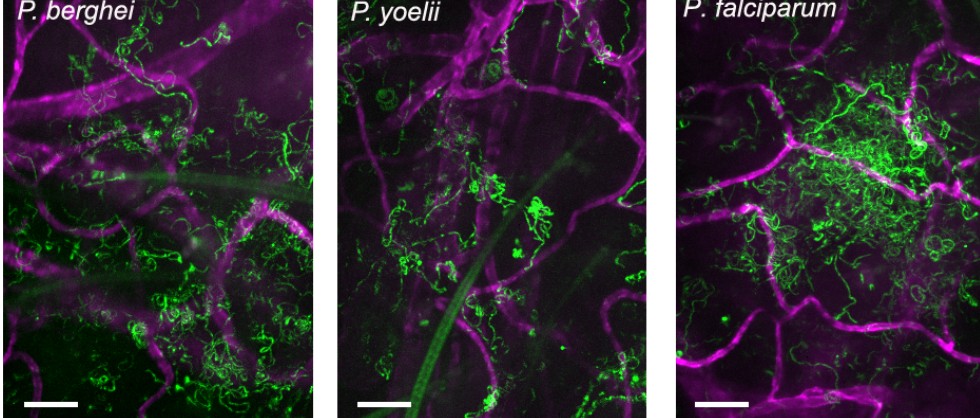

**Figure 1. *P. berghei*, *P. yoelii*, and *P. falciparum* sporozoites moving in the dermis.**

Time-lapse microscopy of sporozoites, 10 min after intradermal inoculation into mice with CD31-labeled vascular endothelia. Maximum-intensity projection over 240 s shows trajectories of moving sporozoites (green) and blood vessels (magenta). The light green structures in the first two panels are autofluorescent hair follicles. Scale bar, 50 μm. See Movies EV1–EV3.

**Automated sporozoite detection and tracking**

To streamline quantification of sporozoite motility, we developed an automated method to detect and track sporozoites. Fiji software (Schindelin *et al*, 2012) was used for vigorous background subtraction and thresholding of raw images, and then spot detection and tracking were performed using ICY software (de Chaumont *et al*, 2012). A visual description of the image processing, spot detection and tracking steps can be seen in Fig 2A and B and more detailed information can be found in the Methods. To verify this new method, we compared the data obtained from the automated spot detection and tracking method to our previous

analysis of *P. berghei* motility (Hopp *et al*, 2015), performed through manual sporozoite detection and tracking, using the same set of videos for both types of analysis. Comparing sporozoite speed and displacement obtained with each method showed that overall the automated method generates data comparable to the data obtained with manual sporozoite tracking (Fig 2C and D). These data suggest that the automated tracking method can be used to quantify sporozoite motility in a more high-throughput manner than would be possible with time-consuming manual tracking methods.

Comparing sporozoite speed obtained with both tracking methods showed that the parasite speed does not change significantly

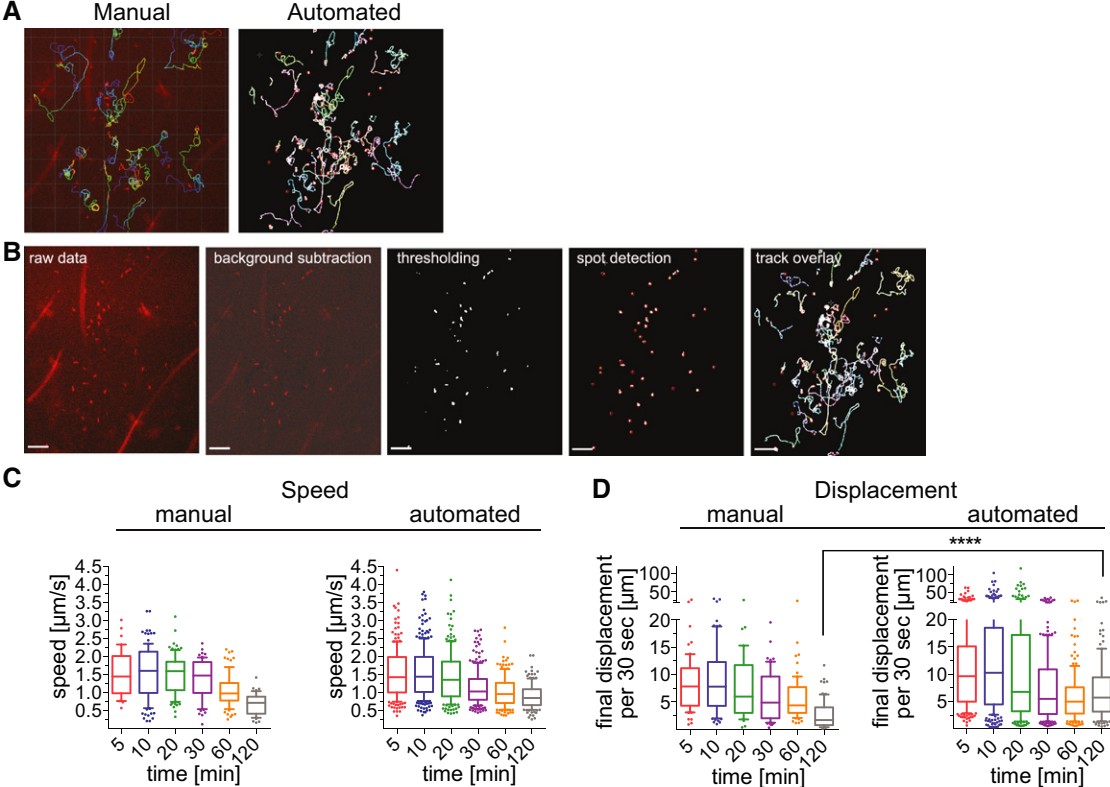

**Figure 2. Automatic tracking of *P. berghei* sporozoites moving in the dermis generates data that is comparable to manual tracking.**

A *P. berghei* sporozoite trajectories obtained with manual (left) and automated (right) tracking methods.

B Image processing and automated tracking method. Shown are images illustrating each step of automated tracking: The raw data image shows the first frame of time-lapse microscopy of *P. berghei* sporozoites (red) after intradermal inoculation, followed by the same image after background subtraction and thresholding using Fiji software. Spot detection was run using ICY software, and the final image shows the overlay with tracks generated using the spot tracking plugin of ICY software. Scale bars, 50 µm.

C Apparent speed of sporozoites 5–120 min after intradermal inoculation, obtained with manual and automated tracking methods. Data are displayed in box and whisker plots with whiskers showing 10–90 percentiles and values below and above the whiskers shown individually. Horizontal line indicates the median speed. No statistically significant differences between the corresponding time points of manual and automated tracking methods were found (Kruskal–Wallis test).

D Displacement of sporozoites 5–120 min after intradermal inoculation, obtained with manual and automated tracking methods. To normalize between tracks of different duration, displacement is displayed as final displacement per 30 s interval. Data are displayed in box and whisker plots with whiskers showing 10–90 percentiles and values below and above the whiskers shown individually. Horizontal line indicates the median. The only statistically significant difference in sporozoite displacement between the manual and automated tracking methods was at 120 min (Kruskal–Wallis test, ****$P < 0.0001$).

Data information: For panels C&D, a varying number of videos were processed for each time point: C: 5 min (3 videos/37 manual tracks/203 automated tracks), 10 min (6 videos/112 manual tracks/292 automated tracks), 20 min (4 videos/95 manual tracks/184 automated tracks), 30 min (6 videos/63 manual tracks/235 automated tracks), 60 min (7 videos/77 manual tracks/162 automated tracks), and 120 min (6 videos/44 manual tracks/151 automated tracks). D: 5 min (3 videos/61 manual tracks/203 automated tracks), 10 min (6 videos/74 manual tracks/292 automated tracks), 20 min (4 videos/47 manual tracks/184 automated tracks), 30 min (6 videos/53 manual tracks/235 automated tracks), 60 min (7 videos/65 manual tracks/162 automated tracks), and 120 min (6 videos/48 manual tracks/151 automated tracks).

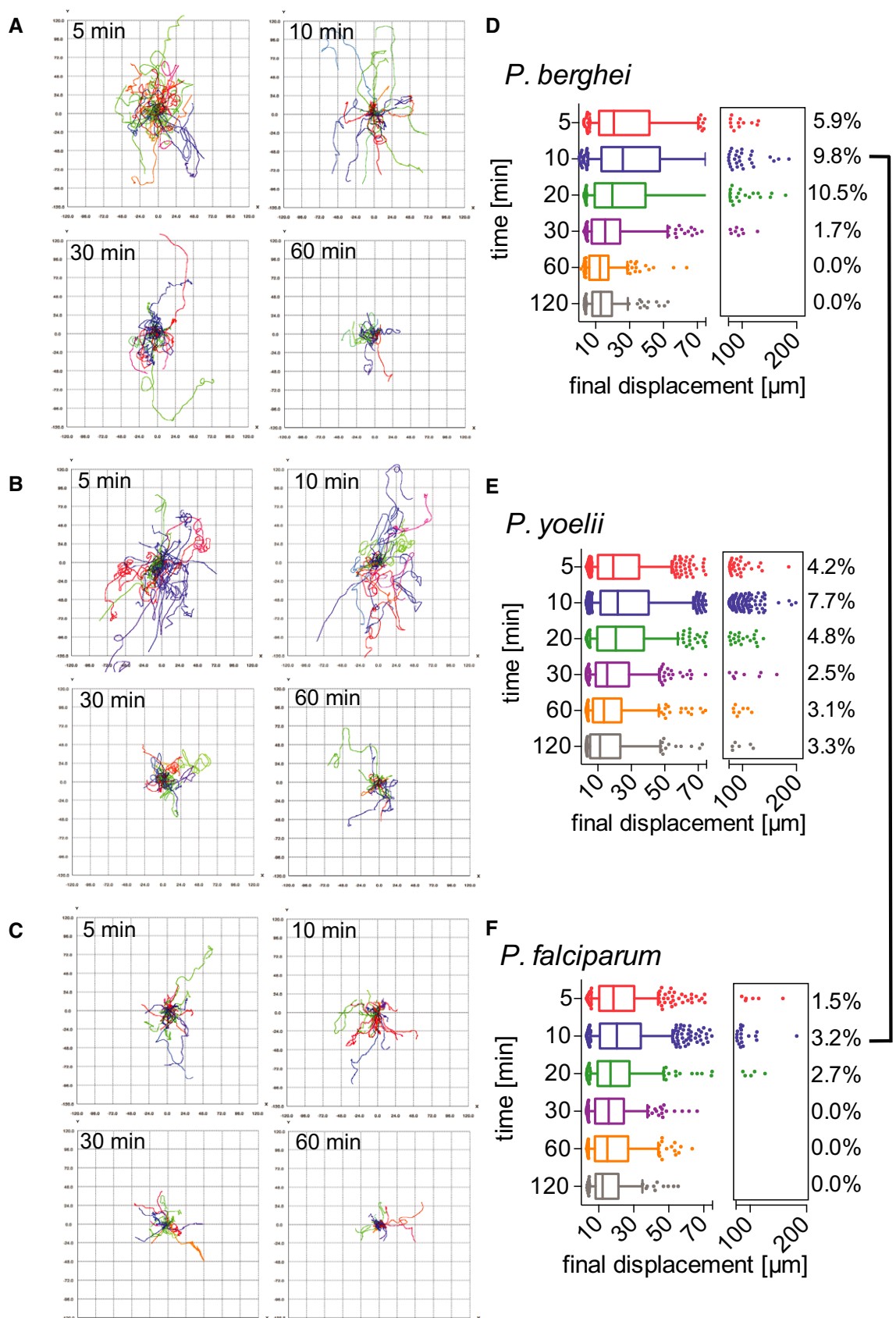

**Figure 3.**

**Figure 3.   Motility of *P. berghei*, *P. yoelii*, and *P. falciparum* sporozoites in the dermis is increasingly constrained over time.**

A–C   Tracks generated through automated tracking were plotted to a common origin to visualize dispersal of *P. berghei* (A), *P. yoelii* (B), and *P. falciparum* (C) at 5, 10, 30, and 60 min after intradermal inoculation. For each time point, the tracks are representative of the average displacement of the entire sporozoite population. Track numbers used for each time point were as follows: 5 min, 40 tracks; 10 min, 32 tracks; 30 min, 25 tracks; 60 min, 21 tracks.

D–F   Displacement of *P. berghei* (D), *P. yoelii* (E), and *P. falciparum* (F) sporozoites 5–120 min after inoculation in box and whisker plots with whiskers displaying the 10–90 percentiles and values below and above the whiskers shown individually. Horizontal lines show median displacement at each time point. Percentage values in boxes indicate the fraction of tracks displacing over 75 micrometers at each time point. Within a given species, the difference in sporozoite displacement between 5 min and 120 min is statistically significant (Kruskal–Wallis test: *P. berghei* $P < 0.0001$, *P. yoelii* $P < 0.0001$, *P. falciparum* $P < 0.01$). Comparisons between species showed no statistically significant difference in displacements for any given time point, with the exception of the 10 min time point in which there was a significant difference between *P. berghei* and *P. falciparum* (Kruskal–Wallis test, *$P < 0.05$). For panels (D–F), a varying number of videos were processed for each time point after inoculation: (D) Same data set as used for Fig 2D: 5 min (3 videos/203 tracks), 10 min (6 videos/292 tracks), 20 min (4 videos/184 tracks), 30 min (6 videos/235 tracks), 60 min (7 videos/162 tracks) and 120 min (6 videos/151 tracks). (E) 5 min (7 videos/714 tracks), 10 min (16 videos/1514 tracks), 20 min (4 videos/415 tracks), 30 min (4 videos/319 tracks), 60 min (4 videos/255 tracks), and 120 min (4 videos/180 tracks). (F) 5 min (6 videos/389 tracks), 10 min (6 videos/464 tracks), 20 min (3 videos/184 tracks), 30 min (2 videos/195 tracks), 60 min (2 videos/239 tracks), and 120 min (2 videos/103 tracks).

over the first 30 min and we find a drop in speed at 120 min by both methods (Fig 2C). No significant differences in sporozoite speed between corresponding time points of the two different tracking methods were found (Kruskal–Wallis comparison; Fig 2C). Nonetheless, some differences in these two data sets are expected, given that the manually tracked data set included only a subset of the total motile sporozoites that were tracked in the automated analysis: While the automated analysis tracks all motile sporozoites, therefore generating tracks of different total durations, the manual tracking analysis only included motile sporozoites that were observed in the field of view for the full duration of the 4 min long video, which was a necessary requirement to allow calculation of mean square displacement (Hopp *et al*, 2015). To be able to directly compare sporozoite displacement for tracks of different durations, displacement was normalized to the average displacement per 30-s interval, thus allowing comparison of the automated and manual tracking data sets. As previously described (Hopp *et al*, 2015), sporozoite displacement drops at 20 min after inoculation in both data sets (Fig 2D). However, automated tracking resulted in statistically higher displacements at the latest time point, 120 min. This is likely due to inclusion of all motile sporozoites in the automated tracking data set, because sporozoites that leave the field of view would be predicted to have higher displacements than those that stay in the field. To determine whether this was the case, sporozoites that were leaving the field of view and thus excluded from the manual analysis, were manually tracked for the 10 min and 120 min time points and added to the previous manual analysis. This showed that the displacement of the total sporozoite population at both 10 min and 120 min after inoculation is higher than was suggested by the original analysis of sporozoites that remain in the field of view throughout the acquired video (Fig EV1).

### Constrained motility of *P. berghei*, *P. yoelii*, and *P. falciparum* sporozoites over time

Using the automated detection and tracking method, we analyzed motility of *P. yoelii* and *P. falciparum* sporozoites moving in the dermis of mice and compared their motility to the data obtained from automated analysis of our previous imaging data of *P. berghei* sporozoites. Sporozoites moving in the dermis were imaged over the first two hours after intradermal inoculation, and videos were acquired at 5, 10, 20, 30, 60, and 120 min after injection (examples shown in Movies EV1–EV3).

Sporozoite trajectories were re-centered to a common origin to visualize sporozoite displacement, showing a gradual decrease in displacement over time for all three *Plasmodium* species (Fig 3A–C), suggesting that similar to *P. berghei*, the displacement of *P. yoelii*, and *P. falciparum* sporozoites decreases over time after dermal inoculation. While the previous displacement analysis for comparison of the automated to the manual tracking data showed displacement per 30-s interval (Fig 2D), this analysis shows the final displacement of motile sporozoites, thus allowing analysis of the total displacement, a parameter that is lost by the normalization to 30-s interval. The analysis of final displacement showed that the median displacement of all three species is comparable, with sporozoites displacing 20 μm, on average, at 5 min after intradermal injection. Interestingly, all three species showed the highest final displacements at 10 min after inoculation (Fig 3D–F). After this time point, final displacement gradually drops to 10–15 μm at 120 min after injection. For each of the three species, the difference in sporozoite displacements between 5 min versus 120 min is statistically significant. Comparisons of displacements between species at each time point were not significant with the exception of the 10 min time point, which was significantly different between *P. berghei* and *P. falciparum*.

We also compared the percentage of sporozoites moving with very high displacements, between 75 and 200 μm (see boxed areas in the right-hand side of the displacement graphs in Fig 3D–F). For all three *Plasmodium* species, the percentage of the population moving these large distances was maximal at 5–20 min. While in the rodent *Plasmodium* species, 9.8% (*P. berghei*) and 7.7% (*P. yoelii*) of sporozoites move this far at the 10 min time point, only 3.2% of *P. falciparum* sporozoites reach displacements this large (Fig 3D–F). Interestingly, while for both *P. berghei* and *P. falciparum*, these high-displacing sporozoites are no longer observed at 60 min after inoculation (Fig 3D and F), *P. yoelii* sporozoites are still moving larger distances, with 3% of sporozoites moving between 50 and 200 μm in displacement at 60 and 120 min after inoculation (Fig 3E). This difference in sporozoite behavior between the two rodent *Plasmodium* species is consistent with previous studies which showed that the infectivity of *P. yoelii* sporozoites to rodents is significantly higher than that of *P. berghei* sporozoites (Weiss, 1990; Khusmith *et al*, 1991; Khan & Vanderberg, 1991; Briones *et al*, 1996) and is in line with the observation of a persistent exit of *P. yoelii* sporozoites out of the dermis into the circulation for over 2 h after inoculation (Yamauchi *et al*, 2007).

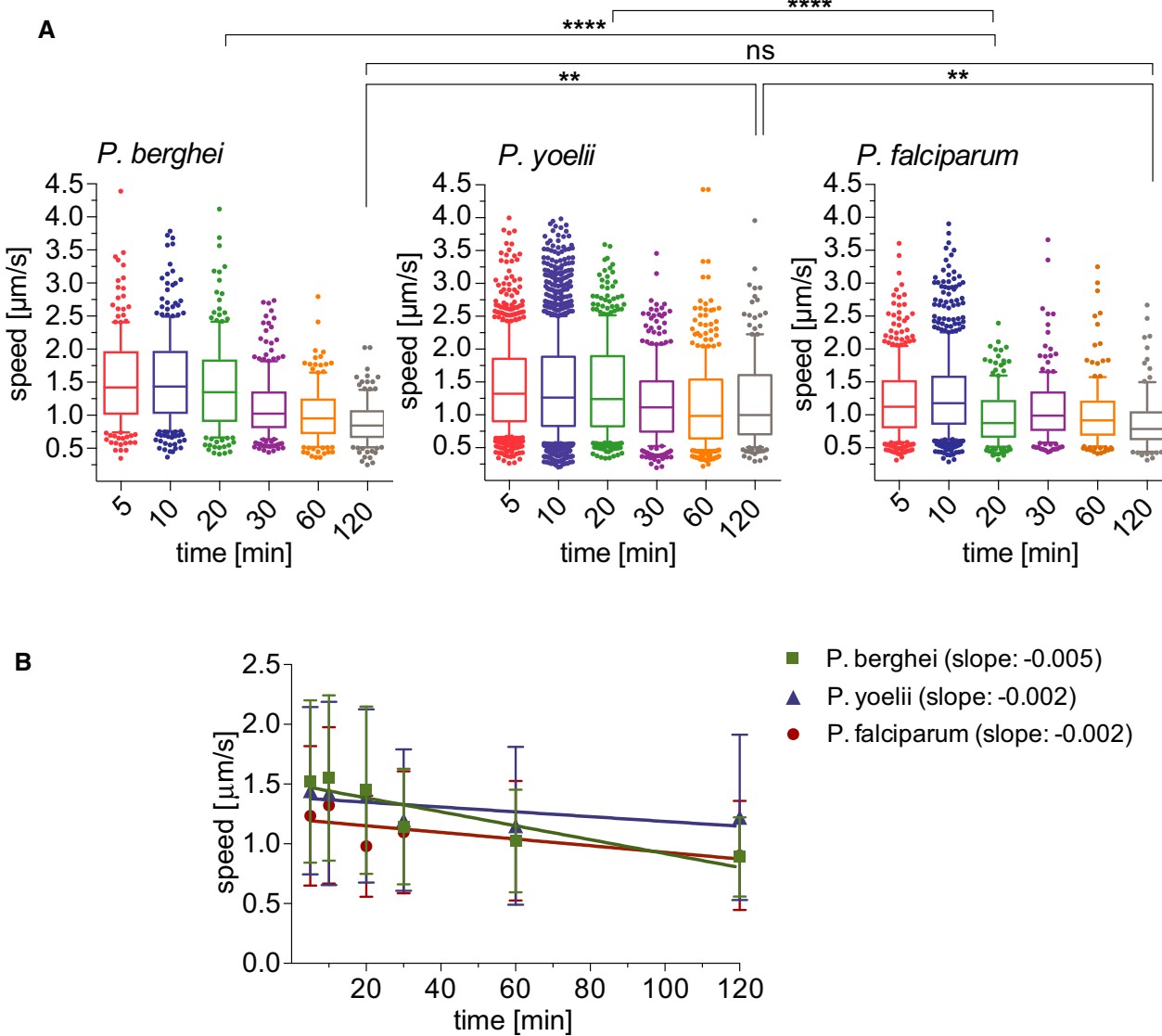

**Figure 4. Apparent speed of *P. berghei*, *P. yoelii*, and *P. falciparum* sporozoites moving in the dermis 5–120 min after intradermal inoculation.**

A   Speed of *P. berghei*, *P. yoelii*, and *P. falciparum* sporozoites 5–120 min after inoculation in box and whisker plots with whiskers displaying the 10–90 percentiles and values below and above the whiskers shown individually. Horizontal lines show the median speed at each time point. Comparisons among the species showed statistically significant difference in sporozoite speeds at the 20 min time point (Kruskal–Wallis test, ****$P < 0.0001$ for both comparisons) and the 120 min time point (Kruskal–Wallis test, **$P < 0.006$ for both comparisons).

B   Linear regression of apparent speed shows a more rapid decrease for *P. berghei* (slope = −0.005) than for *P. yoelii* and *P. falciparum* (slopes = −0.002). Symbols indicate mean values ± SD.

Data information: A varying number of videos were processed for each time point after inoculation, using the same data set used for Fig 3: *P. berghei*; 5 min (3 videos/ 203 tracks), 10 min (6 videos/292 tracks), 20 min (4 videos/184 tracks), 30 min (6 videos/235 tracks), 60 min (7 videos/162 tracks), and 120 min (6 videos/151 tracks). *P. yoelii*; 5 min (7 videos/714 tracks), 10 min (16 videos/1514 tracks), 20 min (4 videos/415 tracks), 30 min (4 videos/319 tracks), 60 min (4 videos/255 tracks), and 120 min (4 videos/180 tracks). *P. falciparum*; 5 min (6 videos/389 tracks), 10 min (6 videos/464 tracks), 20 min (3 videos/184 tracks), 30 min (2 videos/195 tracks), 60 min (2 videos/239 tracks), and 120 min (2 videos/103 tracks).

### High gliding speed of *P. yoelii* sporozoites at late time points

The time-lapse images of *P. berghei, P. yoelii*, and *P. falciparum* sporozoites moving through the dermis of mice were used for analysis of sporozoite gliding speed. Of note, this present analysis of *P. berghei* sporozoite speed was done with a larger set of time-lapse

images than the data set used for the speed analysis of tracks generated by the automated tracking method seen in Fig 2C, which was performed on a smaller data set to match the prior manual analysis (Hopp *et al*, 2015). As Fig 4 shows, sporozoites of all three *Plasmodium* species move at approximately 1.5 μm/s at 5–10 min after inoculation. For *P. berghei*, this speed is stable over the first 30 min and

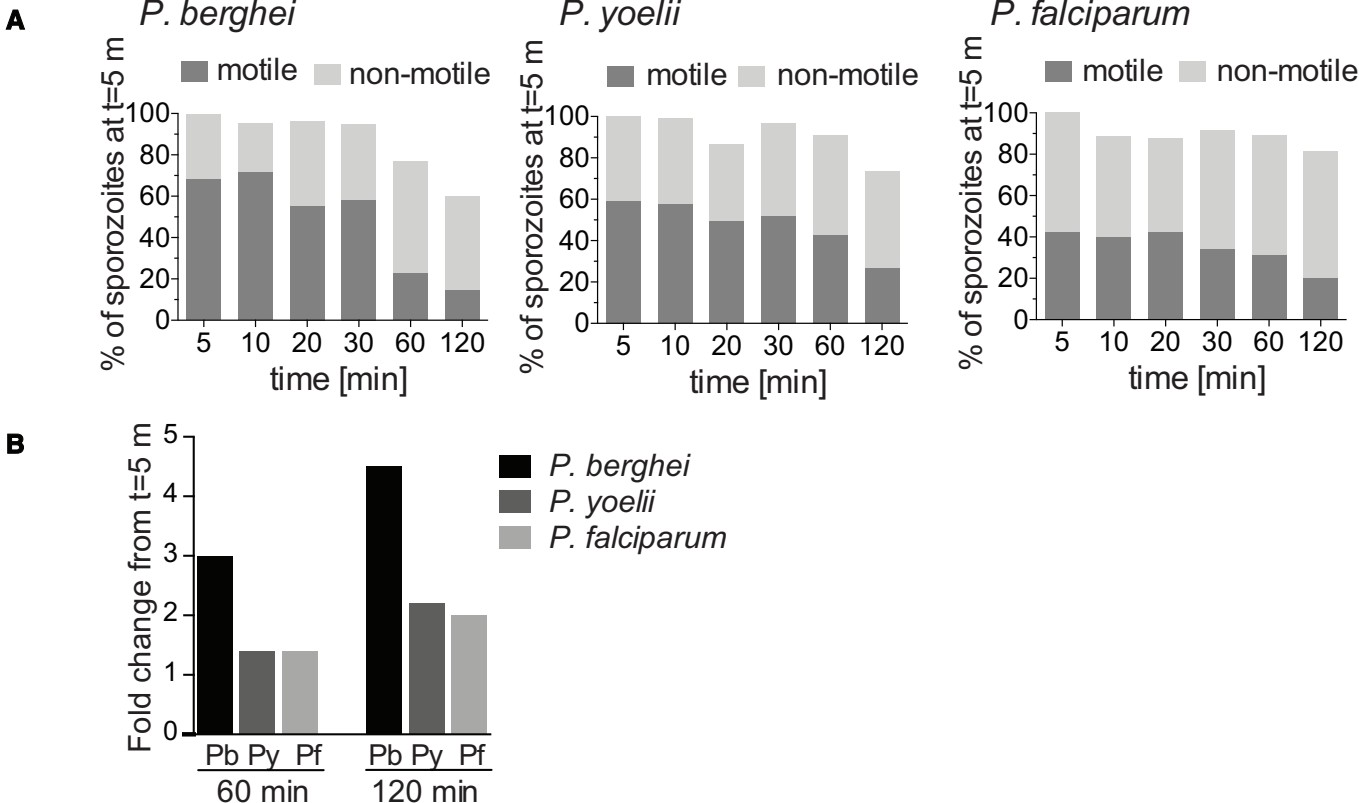

**Figure 5.   Motility frequency over time of *P. berghei*, *P. yoelii*, and *P. falciparum* sporozoites.**

A   The number of motile and non-motile sporozoites moving in the dermis of mice was manually counted in time courses spanning 120 min and is displayed as a percentage of total sporozoites observed 5 min after inoculation, for *P. berghei* (left panel), *P. yoelii* (center panel), and *P. falciparum* (right panel).
B   Fold change in the number of motile sporozoites at 60 and 120 min post-inoculation compared to the number of motile sporozoites at 5 min. The data shown only include complete imaging sessions over 120 min after intradermal injection of sporozoites (*P. berghei*: n = 2, *P. yoelii* n = 4, *P. falciparum* n = 2).

drops off to below 1 μm/s at 60–120 min after inoculation (Fig 4A, left panel), similarly to what was described previously (Hopp *et al*, 2015). In contrast, *P. yoelii* and *P. falciparum* gliding speeds stay more constant over time (Fig 4A, center and right panels), as also shown by linear regression of the apparent speed (Fig 4B). At the 120 min time point *P. yoelii* moves significantly faster than *P. berghei* and *P. falciparum* sporozoites (Fig 4), again consistent with a persistent exit of *P. yoelii* sporozoites out of the dermis for over 2 hours after sporozoite inoculation (Yamauchi *et al*, 2007).

**Motility of *P. berghei*, *P. yoelii*, and *P. falciparum* sporozoites**

As we have described previously, the frequency of motile of *P. berghei* sporozoite motility decreases over time after their inoculation into the dermis (Hopp *et al*, 2015). We investigated whether *P. yoelii* and *P. falciparum* show the same trend in their motility. Videos of time courses were manually counted to determine the number of sporozoites that are motile and non-motile. The data presented include complete time course experiments spanning 5 min to over 120 min after intradermal injection of sporozoites (Fig 5A). To illustrate the percentage of sporozoites leaving the field of view, the data are shown as percentages of the absolute number of sporozoites observed at 5 min after inoculation. We found that

compared to *P. berghei*, which, over the course of 2 h, lose the ability to move in the dermis, the percentage of motile *P. yoelii* and *P. falciparum* does not rapidly drop off and remains similar at later time points. To better illustrate this, we calculated the fold-decrease in motile sporozoites at 60 and 120 min (Fig 5B): The percent of motile *P. berghei* sporozoites drops 3- and 4.5-fold at 60 and 120 min post-inoculation, respectively. In contrast, over the same time frame, the number of motile *P. yoellii* and *P. falciparum* sporozoites decreases 1.4-fold and ~2 fold. This is consistent with the observation of a slow trickle of *P. yoelii* sporozoites out of the dermis over more than 3 hours after inoculation (Yamauchi *et al*, 2007) and suggests that the observed higher infectivity of *P. yoelii* sporozoites compared to *P. berghei* may reflect their biology in the skin in addition to their increased infectivity in the liver (Yamauchi *et al*, 2007; Weiss *et al*, 1990; Khusmith *et al*, 1991; Khan & Vanderberg, 1991; Briones *et al*, 1996).

**Blood and lymphatic vessel entry by *P. berghei*, *P. yoelii*, and *P. falciparum***

Our previous work had shown that *P. berghei* sporozoites interact with CD31[+] blood vessels in the dermis and that while sporozoites are in the vicinity of blood vessels, their motility is more

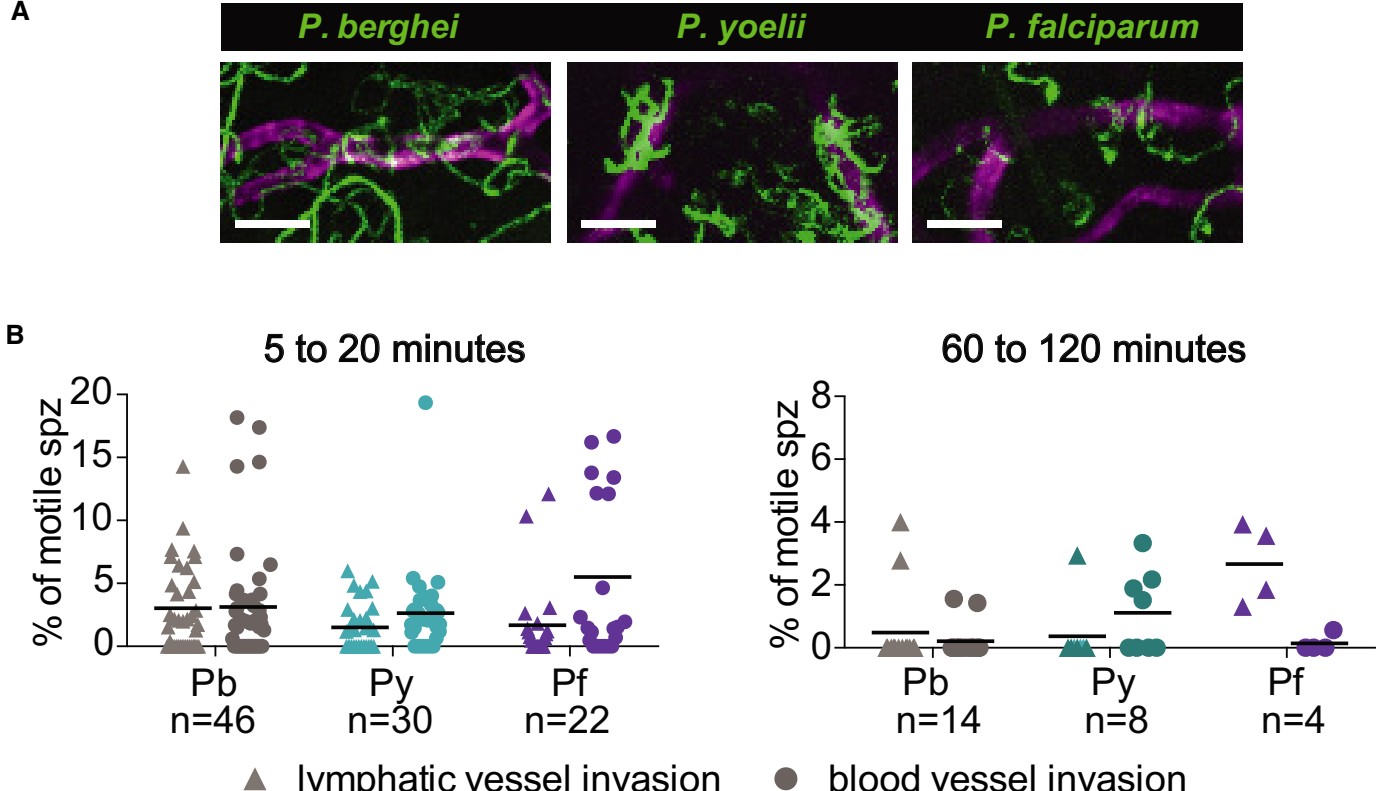

**Figure 6.  P. yoelii and P. falciparum sporozoites engage with and enter blood vessels in skin.**

A    Maximum-intensity projection shows trajectories of moving *P. berghei*, *P. yoelii*, and *P. falciparum* sporozoites (green), engaging with CD31-labeled vascular endothelia (magenta). Scale bars, 25 µm. See Movies EV4–EV6.

B    Lymphatic and blood vessel invasion of *P. berghei*, *P. yoelii*, and *P. falciparum* sporozoites moving in mouse dermis. Combined invasion events observed in 4 min videos recorded at 5, 10, and 20 min after intradermal injection of sporozoites (left panel) and 60 and 120 min after intradermal injection (right panel). *n* = number of videos scored for each *Plasmodium* species. To view a blood vessel entry event by *P. yoelii* or *P. falciparum*, see Movies EV7 and EV8. There were no statistically significant differences in lymphatic or blood vessel entry among the three *Plasmodium* species at 5–20 min after sporozoite inoculation whereas there was a statistically significant increase in lymphatic entry by *P. falciparum* sporozoites compared to *P. berghei* and *P. yoelii* sporozoites at 60 to 120 min after inoculation (Kruskal–Wallis test, *P* < 0.05).

constrained and more circular, likely to optimize contact with dermal capillaries (Hopp *et al*, 2015). To investigate whether *P. yoelii* and *P. falciparum* sporozoites interact with dermal blood vessels in mouse skin, we imaged these sporozoites in conjunction with fluorescently labeled dermal vascular endothelia. We found that similar to *P. berghei*, both *P. yoelii* and *P. falciparum* engage with CD31 blood vessels by frequently circling around the vessel (Movies EV4–EV6). This is illustrated in the maximum projections of sporozoites moving in the vicinity of blood vessels in Fig 6A. We further quantified entry into blood and lymphatic vessels by the sporozoites, classifying these events similarly to what was previously described, with blood vessel entry being defined by a sudden increase in speed and disappearance of the sporozoite out of the field of view and lymphatic entry being defined by the switch from directed forward movement to sideward drifting of the sporozoite at a low velocity (Amino *et al*, 2006; Hopp *et al*, 2015) [see Movies EV7 and EV8 for time-lapse videos of *P. yoelii* and *P. falciparum* blood vessel invasion events; *P. berghei* blood vessel invasion events can be seen in (Hopp *et al*, 2015)]. These events were quantified as a percentage of motile sporozoites in 4 min videos, and data

were pooled from videos acquired at 5, 10, and 20 min after intradermal inoculation, thus covering a total imaging time of 12 min as described previously (Hopp *et al*, 2015). On average 3.1 and 3.0% of *P. berghei* sporozoites exit through the blood vessels and lymphatics, respectively, during 4 min videos recorded at 5- to 20-min post-sporozoite inoculation (Fig 6B, left panel). Both *P. yoelii* and *P. falciparum* sporozoites were found to show similar exit rates (Fig 6B, left panel): We found that 2.6% of *P. yoelii* sporozoites exit through blood vessels, while *P. falciparum* sporozoites were found to enter blood vessels at a slightly higher rate than the rodent parasites, with 5.5% of *P. falciparum* sporozoites entering blood vessels during the acquired imaging time. Both *P. yoelii* and *P. falciparum* sporozoites showed a lower rate of lymphatic invasion (1.5 and 1.7%, respectively) compared to *P. berghei*. There were no statistically significant differences in the rate of blood or lymphatic vessel entry among the three *Plasmodium* species.

Given that *P. yoelii* showed more constant gliding speed throughout the first two hours after inoculation and a higher number of sporozoites displacing large distances at 60–120 min after inoculation, we decided to investigate whether this translates to blood

vessel entry at these later time points. Interestingly, while for *P. berghei*, the rate of blood vessel invasion at these later time points is significantly reduced compared to 5–20 min after inoculation (Kruskal–Wallis test, $P = 0.0301$), this was not the case for *P. yoelii*, where even at 60 and 120 min after inoculation sporozoites were still invading blood vessels (Fig 6B, right panel), consistent with the observed higher infectivity of *P. yoelii* sporozoites to rodents compared to *P. berghei* (Weiss, 1990; Khusmith *et al*, 1991; Khan & Vanderberg, 1991; Briones *et al*, 1996).

### *P. falciparum* sporozoite motility in the dermis of a humanized mouse model

Thus far, our investigations demonstrate that *P. falciparum* sporozoites move and enter blood vessels in mouse skin. To more thoroughly investigate the nature of *P. falciparum* sporozoite motility *in vivo*, we developed an *in vivo* humanized mouse model of *P. falciparum* skin infection. This was accomplished by grafting human neonatal foreskin onto immunocompromised NOD/severe combined immune deficiency (*scid*) gamma (NSG) mice, which can readily accept human skin grafts (Chudnovsky *et al*, 2005). Importantly, the grafted human skin retains the human vasculature and the blood supply to the graft is restored through spontaneous anastomosis of murine and human microvessels (Murray *et al*, 1994; Ho *et al*, 2000). At 4 weeks after grafting the human skin onto the NSG mice, fluorescent labeling of the human vasculature *in vivo* by intravenous injection of anti-human CD31 (which specifically recognizes human endothelial cells) confirmed that many of the blood vessels in the grafted human skin were of human origin (Fig 7A).

*Plasmodium falciparum* sporozoites were injected intradermally into the grafted human skin and intravital imaging performed as previously described (Ho *et al*, 2000). *Plasmodium falciparum* sporozoites were motile and moved through the grafted human skin tissue. Nonetheless, unlike in the rodent dermis, where at 5 and 10 min after inoculation the mean displacement of the *P. falciparum* sporozoite population was 22.5 μm and 25.33 μm, respectively; in the human skin graft, the mean displacement at these same time points were 18.6 μm and 16.6 μm, respectively. Indeed, comparison of *P. falciparum* sporozoite displacement in mouse tissue versus the human xenograft was significantly lower in the xenograft at all corresponding time points (compare Figs 7B and 3F: Kruskal–Wallis comparisons: 5 min: $P < 0.05$; 10 min: $P < 0.0001$; 20 min: $P < 0001$; 30 min: $P < 0.005$; 60 min: $P < 0.0001$). Additionally, *P. falciparum* sporozoites moved with significantly reduced speed at most time points in human skin compared with what was observed in mouse skin (compare Figs 7C and 4A right panel: Kruskal–Wallis comparisons: 5 min: $P < 0.002$; 10 min: $P < 0.0001$; 20 min: $P < 0.0001$; 30 min: ns; 60 min: $P < 0.02$).

The reduced displacement prompted us to analyze the track straightness in order to determine whether these sporozoites were moving in more constrained paths or simply not moving large distances, both of which would result in decreased displacements. We hypothesized that in the human skin grafts, the anastomoses between human and rodent vessels could result in larger, more tortuous vessels, such that sporozoites in the grafted skin could be in closer proximity to vessels and thus be displaying the more constrained motility we had previously observed when *P. berghei* sporozoites were near vessels (Hopp *et al*, 2015). We therefore

measured track straightness, the ratio of displacement to track length (Beltman *et al*, 2009), to quantify the confinement of motility in grafted human skin and mouse skin. If sporozoite tracks are more constrained, the ratio will be smaller, while straight sporozoite trajectories would result in a larger ratio. This analysis showed that indeed *P. falciparum* sporozoite track straightness in the human skin graft is significantly reduced, compared with the track straightness in mouse skin (Fig 7E), indicating that *P. falciparum* sporozoites move in more constrained circular paths in the grafted human skin.

To determine whether this was due to the human skin environment or to the blood vessels that form in the graft, experiments were performed on human skin specimens *ex vivo* prior to grafting onto mice (Fig EV2). Though these specimens do not have a functioning blood supply, the tissue remains alive, being sustained by the oxygen and glucose in the surrounding medium. Interestingly, in the human skin *ex vivo*, *P. falciparum* sporozoite displacement was similar to that in mouse skin and significantly higher than that observed in the human skin xenograft (compare Figs 7B and EV2 panel A: Kruskal–Wallis comparisons: 5 min $P < 0.05$; 10 min $P < 0.0001$; 20 min $P < 0001$; 30 min $P < 0.0001$; 60 min $P < 0.0001$). We also compared track straightness of sporozoite trajectories in human skin *ex vivo* to grafted human skin. Interestingly, sporozoite tracks in human skin *ex vivo* were significantly less confined than those in the grafted human skin, similar to what we observed in mouse skin (Fig 7E). Thus, the lower average displacement of *P. falciparum* sporozoites in human skin xenografts appears to be specific to grafted human skin that possess a blood supply.

We next quantified the number of motile sporozoites over time, manually counting motile and non-motile sporozoites and expressing these data as a percentage of the total sporozoites observed at the 5 min time point. Interestingly, over 60% of inoculated *P. falciparum* sporozoites were motile in the grafted human dermis (Fig 7D), compared with the 40% *P. falciparum* motile sporozoites in rodent dermis (Fig 5A). Moreover, in the grafted human skin, the number of *P. falciparum* sporozoites leaving the field of view was increased compared with rodent dermis, as only ~50% of sporozoites remained at 120 min after inoculation. This is likely due to the higher number of motile sporozoites and the larger percentage of *P. falciparum* sporozoites entering blood vessels. As shown in Fig 7F, more than 10% of motile *P. falciparum* sporozoites enter blood vessels in the human skin graft compared with 5.5% of motile *P. falciparum* in mouse skin. Note that there was no lymphatic invasion in the skin graft since lymphatic vessels are not reconstituted after surgery (Nakamura *et al*, 2018). This higher level of blood vessel entry may be due to the somewhat abnormal anatomy of the blood vessels in the grafted skin. Indeed, comparison of the blood vessels in the engrafted skin (Movies EV9–EV11) with those of normal mouse skin (Movies EV1–EV8), show areas of widening and irregular shaped vessels in the graft.

### Reduced motility of *P. falciparum* sporozoites in the skin of passively immunized mice

Lastly, we wanted to determine whether quantification of *P. falciparum* sporozoite motility *in vivo* can serve as a read-out for efficacy of antibodies targeting sporozoites. We used mouse skin for these assays rather than human skin xenografts because the higher speeds

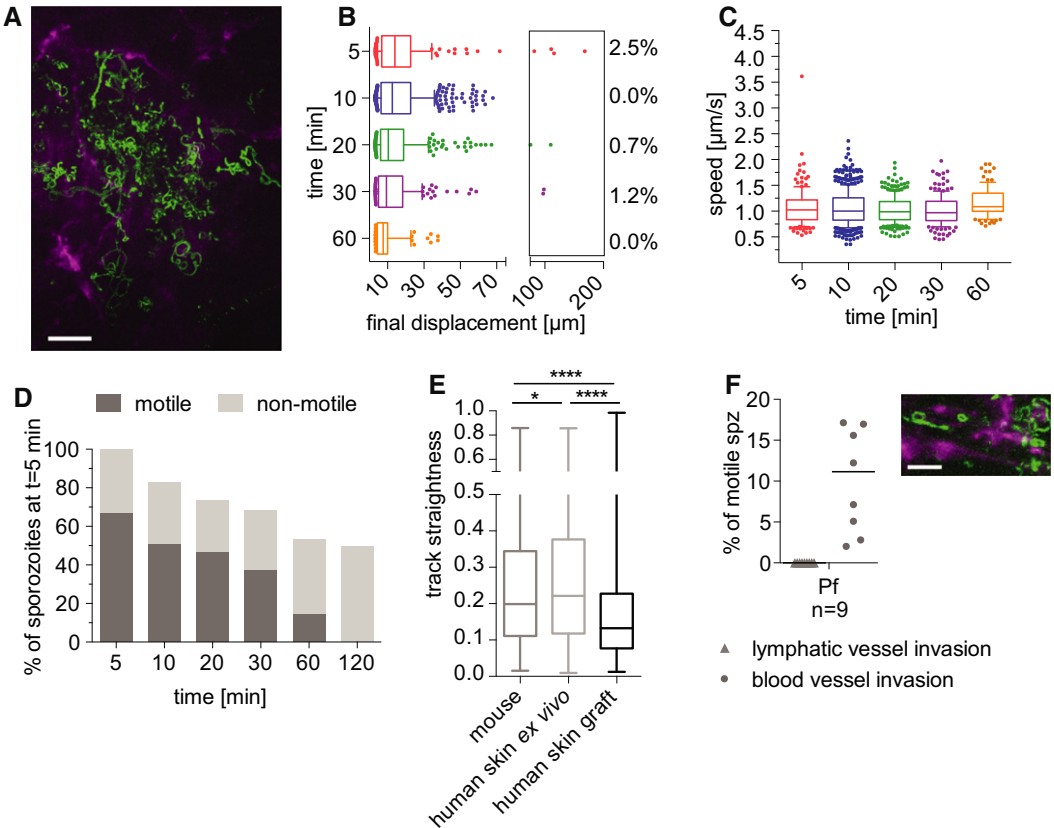

**Figure 7.  Time-lapse microscopy of *P. falciparum* sporozoites moving in human dermis grafted on NSG mice.**

A   Maximum-intensity projection of video acquired 10 min after inoculation of *P. falciparum* sporozoites visualizes trajectories of parasites, with CD31-labeled vascular endothelia (magenta). Scale bar, 50 μm. See Movie EV9.

B   Displacement of sporozoites 5–60 min after inoculation. Data are displayed in box and whisker plots with whiskers showing the 10 to 90 percentiles and values below and above the whiskers shown individually. Horizontal lines show the median at each time point. Percentage values indicate the fraction of tracks displacing over 75 μm.

C   Apparent speed of sporozoites 5–60 min after inoculation. Data are displayed in box and whisker plots with whiskers showing 10–90 percentiles and values below and above the whiskers shown individually. Horizontal lines show the median speed at each time point.

D   Proportion of motile and non-motile sporozoites was manually counted and is displayed as percentage of sporozoites observed 5 min after inoculation. The data shown were obtained from one complete imaging session over 120 min after intradermal injection of sporozoites. See Movie EV9.

E   Sporozoite track straightness, the ratio of displacement to track length of *P. falciparum* sporozoites moving in mouse dermis, human skin *ex vivo*, and human skin graft, at 5- to 10-min post-intradermal injection. Data are displayed in box and whisker plots with horizontal line showing the median and the whiskers extending to the minimum and maximum values. Comparisons showed statistically significant differences in track straightness between the human skin graft and both mouse skin (****$P = 0.0001$) and human skin *ex vivo* (****$P < 0.0001$) using Tukey's multiple comparisons test. A varying number of tracks were processed for each environment: mouse skin: 17 videos, 822 tracks; human skin *ex vivo*: 12 videos, 761 tracks; human skin graft: 9 videos, 718 tracks.

F   *Plasmodium falciparum* sporozoites engage with and enter blood vessels in human skin graft. Maximum-intensity projection shows trajectories of moving *P. falciparum* sporozoites (green), engaging with human CD31-labeled vascular endothelia (magenta). Scale bar, 25 μm. See Movie EV10. Graph shows lymphatic and blood vessel entry events of *P. falciparum* sporozoites in human skin graft. Combined invasion events observed at 5–20 min after intradermal injection in *n* number of videos. To view a blood vessel invasion event of *P. falciparum* in human skin graft, see Movie EV11.

Data information: For panels B–C, data shown originate from the analysis of 3–4 videos per time point and a varying number of tracks were processed for each time point after inoculation: 5 min (157 tracks), 10 min (568 tracks), 20 min (286 tracks), 30 min (171 tracks), and 60 min (101 tracks). No motile sporozoites were observed 120 min after inoculation.

and displacements in mouse skin are likely more relevant to *P. falciparum* motility *in vivo*. We first tested the well-characterized monoclonal clone 2A10, specific for the *P. falciparum* CSP repeat region (Nardin *et al*, 1982), which has been shown to inhibit infectivity of *P. falciparum* sporozoites *in vitro* (Hollingdale *et al*, 1984). To determine its impact in our *in vivo* assay, we inoculated 150 μg of mAb 2A10 intravenously into mice 16–24 h prior to inoculation of sporozoites. Ten minutes after *P. falciparum* sporozoite inoculation, we quantified the number of motile sporozoites, gliding speed, and

displacement. As shown in Fig 8A and B, all of these motility parameters were significantly reduced in passively immunized animals. Importantly, blood vessel entry events were also significantly reduced in mAb 2A10 inoculated mice compared to controls. We then tested mAb 10 and CIS43, two human monoclonal antibodies specific for *P. falciparum* CSP, which have recently been shown to inhibit *P. falciparum* sporozoite infectivity in a variety of assays, though neither was previously tested for their impact on sporozoite motility (Kisalu *et al*, 2018). mAb 10 binds to the CSP repeats and

CIS43 is a bifunctional antibody binding to the repeats and an upstream sequence adjacent to a conserved proteolytic cleavage site (Coppi *et al*, 2005, 2011; Kisalu *et al*, 2018). When tested in our intravital imaging assay with *P. falciparum* sporozoites, we found that both mAb 10 and CIS43 decreased the number of motile sporozoites and decreased blood vessel entry, however, not to the same degree as mAb 2A10 and not reaching statistical significance (Fig 8C). This could be due to the fact that CIS43 and mAb 10 are human mAbs, which may be cleared more rapidly from the mouse's blood circulation leading to lower antibody concentrations in the skin, a possibility that deserves further investigation. Furthermore, in a recent publication, CIS43, was found to be a more potent inhibitor of *P. falciparum* sporozoite infectivity than mAb 2A10 (Kisalu *et al*, 2018) because of its bifunctionality, which enables it to inhibit sporozoites both at the inoculation site, where it acts on sporozoite motility (Fig 8C) and in the liver, where it inhibits proteolytic processing of CSP, which is required for hepatocyte infection (Coppi *et al*, 2005, 2011; Kisalu *et al*, 2018). Our assay tests for sporozoite motility and blood vessel entry at the inoculation site. Importantly, the isotype control human mAb VRC01 (Wu *et al*, 2010) did not have inhibitory activity (Fig 8C). Overall, our data suggest that intravital imaging of *P. falciparum* sporozoites in mouse skin can provide valuable information on the inhibitory activity of antibodies at the dermal inoculation site and that this platform can likely be extended to chemical inhibitors targeting motility. A schematic of the steps and time required to test one monoclonal antibody and its relevant control is outlined in Table EV1.

# Discussion

Sporozoite motility is essential for successful localization to and entry into blood vessels at the inoculation site. To date, studies of this phase of infection have focused on the more tractable rodent parasite *P. berghei*. Here we extend these studies to the human malaria parasite *P. falciparum* and to *P. yoelii*, a rodent parasite that is highly infectious to the laboratory mouse. Our comparative analysis enabled the identification of components of sporozoite motility that are conserved across species, as well as behaviors that are species-specific or specific to a particular host–parasite combination. Overall, we find that in contrast to the liver and blood stages of malaria, the skin is not a species-specific barrier to infection, with

our demonstration of normal motility and blood vessel invasion of *P. falciparum* sporozoites in mouse skin. Thus, the first observations of *P. falciparum in vivo* described in this study likely recapitulate the initial phase of malaria infection in humans.

Many components of sporozoite motility described for the rodent species were observed with *P. falciparum* sporozoites moving in rodent dermis. Sporozoite gliding speed and displacement of *P. falciparum* were similar to that of the rodent parasites at all time points after inoculation. Although the percentage of *P. falciparum* sporozoites reaching displacements larger than 50 μm was lower than the respective value for *P. berghei* and *P. yoelii* at each time point, this trend did not reach statistical significance. For all three *Plasmodium* species, maximal displacement was seen 10 min after sporozoite inoculation, suggesting that sporozoites undergo a phase of activation in the dermis. Although the causal factors remain unknown, it may be linked to the finding that exposure to albumin activates sporozoite motility (Vanderberg, 1974). Importantly, *P. falciparum* sporozoites invade blood and lymphatic vessels in mouse skin at a rate comparable to the rodent parasites and were seen to interact with blood vessels in a similar fashion. These data suggest that sporozoite recognition of blood vessels is likely based on molecules that are shared among different host species, however, precisely what sporozoites are sensing is not known. It also indicates that the parasite's motility machinery is sufficiently robust to enable them to enter vessels of different species. Thus, we demonstrate that there are critical cellular and molecular commonalities between mice and primates during the skin phase of the life cycle. The finding that *P. falciparum* sporozoites in mouse skin recapitulate, at least to some degree, canonical sporozoite behavior at the inoculation site, is consistent with a recent study demonstrating *P. falciparum* sporozoites delivered by mosquito bite were infectious in mice with humanized livers (Sack *et al*, 2017). Though the humanized mouse model is an important and powerful model for the study of human malaria parasites, *P. falciparum* sporozoites have a somewhat decreased infectivity in these mice, which could be due to incomplete engraftment of human hepatocytes or to abnormal behavior of sporozoites in mouse skin. Our demonstration that *P. falciparum* behavior at the inoculation site strongly resembles that of the rodent parasites suggests that incomplete engraftment of human hepatocytes likely accounts for this observation and that the way sporozoites move in the skin and recognize blood vessels are conserved features for *Plasmodium* parasites.

---

**Figure 8. Reduced motility of *P. falciparum* sporozoites in mice passively immunized with CSP-specific monoclonal antibodies.**

Mice were injected intravenously with 150 μg of anti-*Pf*CSP or control mouse or human monoclonal antibodies 16–24 h prior to intradermal inoculation with *P. falciparum* sporozoites. Imaging was performed 10 min after sporozoite inoculation.

A  Activity of mAb 2A10 on sporozoite speed and displacement: Data are displayed in box and whisker plots with whiskers displaying the 10–90 percentiles and values below and above the whiskers shown individually. Horizontal lines show the median. For the displacement data, percentage values indicate the fraction of tracks displacing over 75 μm. Statistical analysis: Mann–Whitney, $P = 0.0001$ (speed data) and $P = 0.0098$ (displacement data). Data shown originate from the analysis of 5 videos obtained at 5 separate imaging sessions, which generated a total number of 217 (naïve) and 136 (2A10) sporozoite tracks.

B  Activity of mAb 2A10 on the proportion of motile sporozoites (left panel) and blood vessel entry (right panel). Four-minute videos of mice inoculated with mAb 2A10 and naïve controls were manually counted at 10-min post-*P. falciparum* sporozoite inoculation. Five videos for each condition, from five independent imaging sessions were scored for motility. Four videos per condition from four independent imaging sessions were analyzed for blood vessel entry. Statistical analysis: Percent motile, Mann–Whitney, $P = 0.0317$; blood vessel entry, Fisher's exact test, $P = 0.0163$.

C  Activity of human mAbs 10 and CIS43, specific for *P. falciparum* CSP, and VRC01, an isotype control human mAb specific for HIV-1, on the proportion of motile sporozoites (left panel) and blood vessel entry (right panel). Four-minute videos of mice inoculated with the indicated mAbs were taken 10 min after sporozoite inoculation and manually counted. Data are pooled from 7 biological replicates, naïve and VRC01 ($n = 2$), naïve and mAb 10 ($n = 2$), VRC01 and CIS43 ($n = 3$). No comparisons were statistically significant (Mann–Whitney for percent motile and Fisher's exact test for blood vessel entry).

---

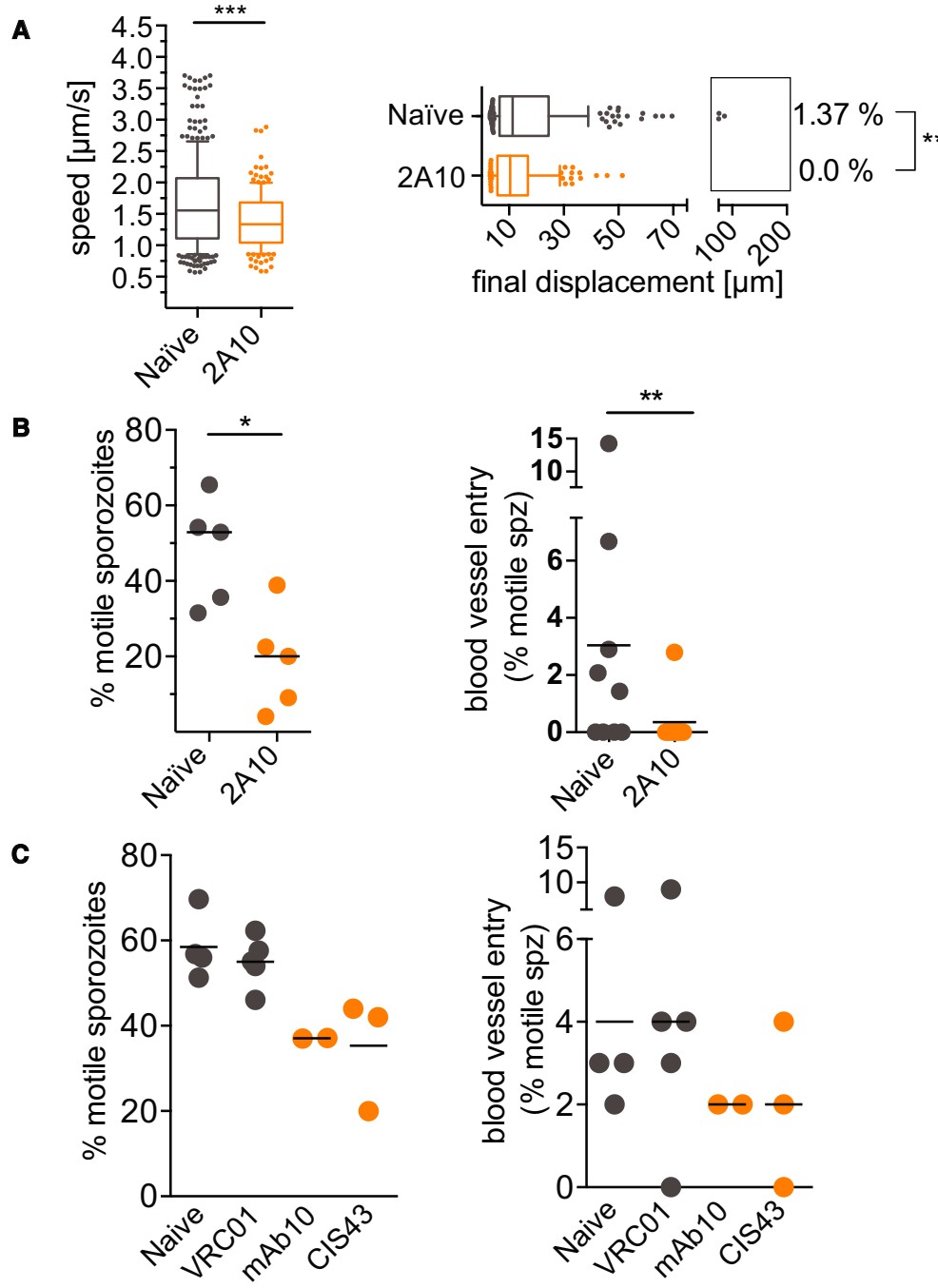

**Figure 8.**

Though the behavior of human and rodent malaria sporozoites in mouse skin share many features, we also observed differences that were host-specific. For both rodent parasites, the proportion of the inoculum that is motile within 5–10 min of inoculation was approximately 70 and 55% for *P. berghei* and *P. yoelii* sporozoites, respectively. In contrast, approximately 40% of *P. falciparum* sporozoites were motile shortly after their inoculation and this percentage remained fairly consistent throughout the 60-min observation period. The continued motility of *P. falciparum* sporozoites over 60 min is similar to *P. yoelii*, whereas the proportion of *P. berghei*

sporozoites, that continue to be motile over a 60-min timeframe, significantly decreases. However, the overall lower percentage of motile *P. falciparum* sporozoites suggests that there may be some host-specific signals that enhance motility after inoculation into the dermis. Interestingly, when inoculated into the grafted human skin these differences in *P. falciparum* sporozoite motility disappear: A higher proportion of the inoculum is motile, with over 60% of sporozoites motile at 5 min after inoculation, suggesting that there are indeed host-specific components to sporozoite motility in the skin. Nonetheless, we found that there are some limitations to the

humanized mouse model. Foremost are the lower displacements and speed of *P. falciparum* sporozoites observed in the human skin graft. We think this is due to the anastomoses between human and rodent vessels, which result in larger and more tortuous vessels. Given that these vessels are larger, it is likely that sporozoites in the grafted human skin were in closer proximity to vessels, and this might explain the more constrained motility and lower speed of the *P. falciparum* sporozoites in human skin compared with the mouse skin. Interestingly, speed and displacement of *P. falciparum* in human skin *ex vivo* were similar to what we observed in mouse skin, further supporting this hypothesis. Thus, it is likely the anatomy of the grafted and anastomosed vessels in the human skin graft that is responsible for the decreased displacement and speed of sporozoites in this setting. Consistent with these observations is our finding of increased blood vessel entry in the skin grafts. Though the human skin xenograft model is being increasingly used in skin physiology and disease studies (reviewed in (Salgado *et al*, 2017; Youn *et al*, 2020)), the nature of the anastomosing blood vessels remains poorly understood.

Comparison between the two rodent malaria species, both of which are commonly used for vaccine studies, demonstrated that *P. yoelii* sporozoites are able to move more persistently, with higher speed and larger displacements at later time points. We also observed more blood vessel invasion events by *P. yoelii* sporozoites at later time points. These data suggest that *P. yoelii* sporozoites have a longer period of time during which they are infectious post-inoculation. In contrast, *P. berghei* sporozoite motility and blood vessel entry decrease significantly at later time points. This could be explained by a more rapid switch to a hepatocyte-invasion mode in *P. berghei*, a possibility that is consistent with its more promiscuous infectivity *in vitro* (Silvie *et al*, 2007) and could lead to a portion of the *P. berghei* inoculum prematurely switching to an invasion mode in the skin. While previous studies using intravenously inoculated sporozoites demonstrated that *P. yoelii* sporozoites develop better in the liver compared to *P. berghei* (Weiss, 1990; Khusmith *et al*, 1991; Khan & Vanderberg, 1991; Briones *et al*, 1996), our results suggest that the higher infectivity of *P. yoelii* sporozoites is also due to their behavior in the skin, further supporting the notion that sporozoite motility in the dermis is a valid correlate for infectivity of the parasite. Recently, a number of transgenic rodent parasite lines expressing *P. falciparum* or *P. vivax* CSP have been developed for pre-clinical vaccine testing in the rodent model (Porter *et al*, 2013; Espinosa *et al*, 2013; Mizutani *et al*, 2016). The different biology of the two rodent malaria species could impact the interpretation of antibody-based interventions performed in the rodent model and should be considered when evaluating vaccine candidates targeting sporozoites. Given the results of our comparative analysis, a *P. yoelii* parasite expressing *P. falciparum* proteins such as CSP (Zhang *et al*, 2016) may be the more robust model.

The quest for a potent malaria vaccine remains a global priority to decrease clinical disease and mortality and enable elimination. Despite this, the development of a highly effective and durable malaria vaccine has proven challenging. In part, this goal has been hampered by the lack of robust pre-clinical models for vaccine testing. Recent efforts have focused on characterizing the parasite-specific antibody response in vaccinated or naturally exposed individuals (reviewed in refs. Cockburn & Seder, 2018; Tan *et al*, 2019), which has opened up new possibilities for epitope-focused vaccine design

and passive antibody therapies. Here we describe an *in vivo* assay in mice, which can be used to screen human monoclonal antibodies and vaccine candidates for their efficacy in targeting *P. falciparum* sporozoites. Indeed, we validate this model using three inhibitory monoclonal antibodies specific for *P. falciparum* CSP. This is a significant step forward and has several advantages over currently used assays for pre-clinical testing of inhibitory activity on *P. falciparum* sporozoites. *In vitro* assays quantifying cell traversal and invasion suffer from the low infectivity of *P. falciparum* sporozoites *in vitro* and our lack of knowledge on how to activate *P. falciparum* sporozoites for these processes once they have been dissected from mosquito salivary glands. A recent study visualized *P. falciparum* sporozoites in human skin explants and developed an automated tool to track their movements (Winkel *et al*, 2019), yet the lack of blood supply to the tissue limits the informative observation period and does not allow for the assessment of blood vessel entry, a critical component of infectivity. Significant progress has been made with the development of a humanized mouse model (Minkah *et al*, 2018); however, the mice used for these assays are expensive and immunocompromised, making it difficult to evaluate vaccine candidates. Our model, using immunocompetent mice, enables the use of larger numbers of mice due to their low-cost relative to the humanized mice, and evaluation of vaccine candidates due to their fully competent immune system. Our model could also be used to evaluate compounds for their inhibitory effect on *P. falciparum* sporozoite motility *in vivo*. *In vitro* screens with the rodent parasite *P. berghei*, such as the one performed by Douglas *et al* (2018) identified several inhibitory compounds of motility, which could now be further tested on *P. falciparum* sporozoites *in vivo*. Moreover, the automated sporozoite tracking tools described in this paper and in the study by Winkel *et al* (2019) facilitate more rapid data analysis. Together this makes for a platform that will enable higher throughput testing of vaccines, chemical inhibitors and antibodies targeting human malaria sporozoites, prior to costly primate or human studies.

# Materials and Methods

All animal experiments were approved by the Johns Hopkins Animal Care and Use Committee.

### Generation of fluorescent *Plasmodium yoelii* –mCherry

The targeting plasmid pL1849-mCherry$_{CSP}$, containing sequences that would target the *mcherry* transgene to the *P. yoelii 230p* locus, and the *mcherry* coding sequence flanked by the *P. berghei csp* 5'UTR and the *P. berghei dhfr* 3'UTR was generated as follows: the *5'pbcsp-mCherry-3'pbdhfr* cassette was amplified from plasmid pL0047 (RMgm database http://www.pberghei.eu), with primers pL0047-PbCS-mCherry-F and pL0047-PbCS-mCherry-R (see Appendix Table S1 for primer sequences), which added XmaI sites for cloning into pL1849 (Lin *et al*, 2011), which contained the *P. yoelii 230p* locus (PY03857) 5'- and 3'-targeting sequences (Appendix Fig S1). The resulting construct pL1849-mCherry$_{CSP}$ was released from the plasmid through digestion with BstXI and ScaI and used for transfection into the *P. yoelii* line Py17X$_{GIMO}$ (line 1923cl1; Lin *et al*, 2011), according to standard procedures (Janse

et al, 2006). The following day, negative selection with 1 mg/ml 5-FC in drinking water (Orr et al, 2012) was used to enrich for parasites that integrated the *PbCSP-mCherry-3'pbdhfr* cassette and excised the *hdhfr*::*yfcu* locus through double cross-over homologous recombination. The resulting PymCherry line was cloned and diagnostic PCRs confirmed integration and absence of the h*dhfr*::y*fcu* marker (Appendix Fig S1). See Appendix Table S1 for all primer sequences. Clone D2 was used for all experiments in the present study.

## Intravital imaging

Sporozoites inoculated into the ear pinna of mice were imaged as described previously (Amino et al, 2007; Hopp et al, 2015). Briefly, anesthesia of 4- to 6-week-old C57BL/6 mice from Taconic was done by intraperitoneal injection of ketamine/xylazine (35–60 μg ketamine/gram body weight), and sporozoites were injected intradermally into the ear pinna in a total volume of 0.2 μl using a NanoFil-10-μl syringe with a NF33BV-2 needle (World Precision Instruments). All sporozoite injections were completed within 30 min of salivary gland dissection to ensure that imaging and parasite conditions remained constant. No purification or centrifugation of sporozoites was performed as we found that these can significantly impact sporozoite viability. The ear pinna was taped to a coverslip, and the mouse was mounted on the platform of an inverted Zeiss Axio Observer Z1 microscope with a Yokogawa CSU22 spinning disk and a preheated temperature-controlled chamber at 30°C. Parasites were imaged with a 10 × objective and magnified using a 1.6 Optovar, resulting in an x and y imaging volume of 500 μm × 500 μm. Stacks of 3–5 slices spanning a total depth of 30–50 μm were imaged. Exposure time per slice was 100 to 150 ms, and stacks were captured at approximately 1 Hz over a total time of 4 min using an EMCCD camera (Photometrics, Tucson, AZ, United States) and 3i slidebook 5.0 software. Indicated time points after inoculation refer to the starting time of the 4-min acquisition. In order to visualize dermal blood vessels, mice were intravenously injected with 15 μg of rat anti-mouse CD31 coupled to Alexa-Fluor-647 (clone 390, BioLegend, San Diego, CA, United States) approximately 2 h prior to imaging. For experiments testing the impact of antibody on sporozoite motility, 150 μg of antibody (in 200 μl of PBS) was inoculated intravenously 16–24 h prior to sporozoite inoculation. Antibodies used in this study were mouse mAb 2A10 specific for the repeat region of *P. falciparum* CSP (Nardin et al, 1982) and human mAbs VRC01 an isotype control specific for HIV-1 (Wu et al, 2010), mAb 10 specific for the repeat region of *P. falciparum* CSP (Kisalu et al, 2018) and CIS43 a bifunctional mAb that binds to both the repeat region of *P. falciparum* CSP and a junctional epitope immediately upstream of the repeats (Kisalu et al, 2018).

## Human skin graft

Each 6- to 12-week-old female NOD *skid* gamma (NSG) mouse (NOD. Cg-Prkdc^scid Il2rg^tm1Wjl/SzJ, Jackson Laboratories) was anesthetized with inhalation isoflurane (2%), and their entire back and bilateral chest walls were shaved with clippers. To make a recipient wound bed for human donor foreskin engraftment, the skin was sterilized with alcohol (70%) and povidone–iodine (Betadine) and survival surgery was performed to excise a 1.5 × 1.2 cm rectangular

area of full-thickness skin below the shoulder blades. Each normal human foreskin specimen was obtained from discarded male newborn foreskin at the Johns Hopkins Hospital and was placed into RPMI at 4°C for 1–4 h prior to use. Each foreskin specimen was cut into a 1.4 × 1.1 cm rectangular area and the subcutaneous fat and fascia sterilely removed with scissors and forceps. Each foreskin graft was placed onto the wound bed on each recipient NSG mouse and secured with interrupted nylon sutures between the skin graft and adjacent mouse skin and muscle fascia. Topical bacitracin ointment was applied to the human skin graft site, and the graft was then covered with a non-adherent dressing (Telfa, Covidien) followed by an adhesive bandage (Band-aid, Johnson & Johnson). Each NSG mouse with engrafted human foreskin was singly housed in an autoclaved cage with water supplemented with 0.5 mg/ml enrofloxacin (Baytril) and aspartame sweetener. After 2 weeks, bandages and sutures were removed, and mice were monitored for another 2 weeks to ensure graft survival prior to using the mice in subsequent experiments.

## Imaging of sporozoites in human skin graft

Anesthesia of NSG mice with human skin graft was done by intraperitoneal injection of ketamine/xylazine (35–60 μg ketamine/gram body weight). Similar to a previously described protocol (Ho et al, 2000), a midline dorsal incision around three sides of the human skin graft was made without disrupting the lateral dermal blood supply. The skin human graft was separated from the underlying tissue, and the microvasculature within the graft was exposed through dissection of the connective tissue adhered to the graft. *P. falciparum* sporozoites were then injected into the dermis in a total volume of 0.2 μl using a NanoFil-10 μl syringe with a NF36BV-2 needle (World Precision Instruments). The human skin was then moistened with RPMI medium, immobilized to a coverslip with two small drops of superglue and the edges of the human skin graft were covered with lubricant Puralube Vet ointment (Dechra, NDC 17033-211-38) to avoid drying out of the tissue. The mouse was mounted on the platform of an inverted Zeiss Axio Observer Z1 microscope with a Yokogawa CSU22 spinning disk and imaging was performed as described above. To visualize human dermal blood vessels, mice were intravenously injected with 15 μg of mouse anti-human CD31 coupled to Alexa-Fluor-647 (clone WM59, BioLegend, San Diego, CA, United States) in 200 μls of PBS, approximately 2 h prior to imaging.

## Automated sporozoite tracking method

The three components of this method: (i) image processing, (ii) spot detection and tracking, (iii) track managing and data export, are described below:

### Image processing
Time-lapse stacks with 3–5 slices spanning a total depth of 30–50 μm were captured at approximately 1 Hz over a total time of 4 min using 3i slidebook 6.0 software (Intelligent Imaging Innovations) and exported as tiff files. Note was taken of the pixel size (pixel size is dependent on camera resolution, using a Photometrics Evolve camera and 16× magnification, the pixel size was 1 μm/pixel) and the time interval, i.e., the time elapsed between two captured Z stacks. Using Fiji software (Schindelin et al, 2012),

which is freely available (https://fiji.sc), the exported raw data as tiff file was opened (File > Import > Image Sequence) and the image was projected into a single Z-layer using a method called maximum-intensity projection (Image > Stacks > Z project > Max Intensity). For background subtraction, an average projection over the Z-dimension was generated (Image > Stacks > Z project > Average Intensity). The resulting average was subtracted from each image of the maximum projected time-lapse series (Process > Image Calculator > Subtract). An example is shown in Fig 2 panel B. The threshold of the time-lapse series was adjusted (Image > Adjust > Threshold > deselect "calculate threshold for each image"), to remove remaining background signal and an example of a thresholded image is shown in Fig 2 panel B.

### Spot detection and tracking

The thresholded time series was analyzed using ICY 1.8.6.0 software (BioImage Analysis Unit; Institut Pasteur; de Chaumont *et al*, 2012) for automated spot detection and track generation. If necessary, the image was converted to a time-lapse series (Sequence Operation > Convert to time). Before spot detection was started, pixel size and time interval were set in "Sequence properties." For spot detection, the Spot Detector plugin was used (Detection & Tracking > Spot Tracking > Run the Spot Detector plugin; Detector > "Detect bright spot over dark background" and select scale 3 (~7 pixels) scale 4 (~13 pixels); Output > select "export to SwimmingPool"). An example of an overlay of detected spots over the original thresholded image is shown in Appendix Fig S2 panel D. The parameters for sporozoite tracking were as follows: expected false detections per frame: 50; probability of detection for each particle: 0.9; expected track length: 50; Minimum probability of existence: 0.5; probability of existence threshold for track termination: 0.0001; single motion model: expected displacement length in the x-y plane: 10; select "use directed motion"; expected displacement length in the x-y plane: 10; select "re-estimate online"; expected number of new objects per frame: 5; expected number of objects in the first frame: 85; depth of the track trees: 4; gate factor for association: 4. These parameters had previously been saved to a configuration file, which was loaded (Spot Tracking > Interface > advanced Interface > Configuration file > load a configuration file). Spot detection results were used for tracking (Spot Tracking > Detection Source > Select detection results here) and the input of spots created in Spot Detection were chosen. An example of an overlay of created tracks over the original thresholded image is shown in Fig 2 panel B.

### Track managing and data export

To remove background noise, tracks of < 30 s in total duration, tracks with a total track length of less than 20 µm, tracks with a total displacement of less than 3 µm and tracks with an average speed higher than 4 µm/s were excluded from the analysis. This was done as follows: Tracks of < 30 s in duration were selected (TrackManager > Edit > Select track by length > 0-X [X = 30,000/interval time]) and removed (TrackManager > Edit > Delete selection). Furthermore, tracks of < 20 µm in total track length were removed (TrackManager > add Track Processor > Motion Profiler > select "Filter tracks..." and "Use real units"; then under "Keep tracks with" select "total displacement > 20" [this removes tracks with a total track length of < 20 µm only if the pixel/ µm ratio is 1]). Also, tracks of less

### The paper explained

#### Problem

*Plasmodium* parasites exhibit a high degree of host specificity, which limits *in vivo* assays with human malaria parasites. Indeed, the generation of a fully efficacious malaria vaccine is hampered by a lack of *in vivo* models for human malaria parasites.

#### Results

After their inoculation by an infected mosquito, *Plasmodium* sporozoites must move in the skin to find and enter blood vessels. Here we use quantitative intravital microscopy to compare the behavior of human and rodent malaria sporozoites at the dermal inoculation site and find that the skin is not a species-specific barrier to infection: Indeed, *P. falciparum* sporozoites move and enter blood vessels in mouse skin similarly to the rodent parasites. Furthermore, we show that passively administered antibodies specific to the major surface protein of *P. falciparum* sporozoites can inhibit their motility and blood vessel entry in the mouse.

#### Impact

Though the skin phase of infection is but one part of the complex life cycle of the malaria parasite, it is a time of demonstrated vulnerability for the parasite. Here we show that we can recapitulate this portion of the human malaria parasite's life cycle in mice and use this to screen antibodies and likely chemical inhibitors prior to expensive human clinical trials.

than 3 µm in final displacement, were removed (TrackManager > add Track Processor > Motion Profiler > select "Filter tracks..." and "Use real units"; then under "Keep tracks with" select "net displacement > 3" [this removes tracks with a total track length of less than 20 µm only if the pixel/ µm ratio is 1]). Speed and displacement data were exported to an Excel file (TrackManager > add Track Processor > Motion Profiler > select "Use real units" > Export statistics) and Track projections to a common origin were created to visualize parasite dispersal file (TrackManager > add Track Processor > Motion Profiler > select "Use real units" > Save graphics in a.png file). An example of a track is shown in Fig 2 panel B. The positional x/y data of all tracks over time were exported (TrackManager > add Track Processor > Track Processor export track to Excel).

### Displacement and speed

Total displacement was calculated as the Euclidean distance between the first and the last sporozoite location. Sporozoite speed was calculated as the total track length, divided by the track duration. This measure is sensitive to the frame rate with which the video was acquired (Kan *et al*, 2014) and to achieve consistent interval times across the entire data set, for the calculation of apparent speed we utilized only videos acquired at 850–1,250 ms per frame.

### Track straightness

The track straightness is defined as the ratio of track displacement to total track length and as a result can range between zero (entirely constrained track) and one (the track is a straight line) (Beltman *et al*, 2009). To calculate the track straightness of *P. falciparum* sporozoite trajectories, the tracks generated for displacement and speed analysis were used.

## Percent motile and blood vessel entry

To quantify the proportion of motile sporozoites and blood vessel entry, videos were manually counted. Sporozoites were considered non-motile if they moved $<\sim 2\ \mu m$ for the duration of the video. Blood vessel entry was defined by a sudden increase in speed or visual entry into the blood vessel, follow by disappearance out of the field of view. When blood vessel entry was compared between antibody inoculated and naïve mice, criteria for inclusion was that at least one blood vessel entry event occurred in either of the two conditions being compared. Lymphatic invasion was defined by the switch from directed forward movement to sideward drifting with low velocity.

## Statistical considerations

The presented data predominantly consists of intravital imaging videos that are subsequently analyzed by the automated tracking method as described above. All videos were analyzed using the same tracking parameters. When quantifying parameters such as percent motile and blood vessel entry, which had to be done manually, each movie was scored on two independent occasions. Mice used for these studies were of the same sex (female), strain (C57Bl/6), vendor, and of similar ages (4- to 6-week-old), with the exception of the mice engrafted with human skin. For this, NSG mice (Jackson) were used; between 10 and 16 weeks of age at the time of the experiment. For each time point with each *Plasmodium* species, with the exception of the 120 min time course of *P. falciparum* in engrafted human skin (Fig 7D), 3–5 biological replicates were performed, where each biological replicate was performed with a different batch of infected mosquitoes and a different mouse.

# Data availability

This study includes no data deposited in external repositories. However, all parasite lines generated for this study are freely available to the scientific community.

Expanded View for this article is available online.

## Acknowledgements
We would like to thank Dr. Godfree Mlambo, Dr. Abhai Tripathi and Chris Kizito, the team of the parasitology and insectary core facilities at the Johns Hopkins Malaria Research Institute for their outstanding work and Bloomberg Philanthropies for their support of these facilities. We also thank Drs. Robert Seder, Azza Idris, and Neville Kisalu for their generous gift of mAbs CIS43, 10, and VRC01. We acknowledge the Johns Hopkins School of Medicine Microscopy Facility (MicFac) and thank Dr. Scott Kuo and Barbara Smith for their invaluable assistance. This work was supported by the National Institutes of Health (R01 AI132359 to PS, R01AR069502 and R01AR073665 to LSM, and K01AR073924 to NKA), by a Johns Hopkins Malaria Research Institute fellowship (CSH and SK) and by Bloomberg Philanthropies.

## Author contributions
Conceptualization: CSH and PS; conducted experiments: CSH, SK, NKA, RJM, HL; supervised experiments: CSH, NKA, LSM, PS; analyzed data: CSH, SK, KKC, PS; wrote manuscript: CSH and PS; edited manuscript: CSH, SK, NKA, RJM, HL, KKC, LSM, PS; funded the work: LSM and PS.

## Conflict of interest

L. S. M. is a full-time employee of Janssen Pharmaceuticals (a Johnson & Johnson Company) and holds Johnson & Johnson stock. L. S. M. performed all work at his prior affiliation at Johns Hopkins University School of Medicine, and he has received prior grant support from AstraZeneca, Pfizer, Boehringer Ingelheim, Regeneron Pharmaceuticals, and Moderna Therapeutics, was a paid consultant for Armirall and Janssen Research and Development, was on the scientific advisory board of Integrated Biotherapeutics and is a shareholder of Noveome Biotherapeutics, which are all developing therapeutics against infections and/or inflammatory conditions.

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
