## [Review Process File · EMBO Molecular Medicine]

Comparative intravital imaging of human and rodent malaria sporozoites reveals the skin is not a species-specific barrier

Christine Hopp, Sachie Kanatani, Nathan Archer, Robert Miller, Haiyun Liu, Kevin Chiou, Lloyd Miller, and Photini Sinnis

DOI: [10.15252/emmm.201911796](https://doi.org/10.15252/emmm.201911796)

Corresponding authors: Photini Sinnis (psinnis1@jhu.edu) , Christine Hopp (christine.hopp@nih.gov)

Review Timeline:

Submission Date:	20th Nov 19
Editorial Decision:	19th Dec 19
Revision Received:	2nd Dec 20
Editorial Decision:	11th Jan 21
Revision Received:	21st Jan 21
Accepted:	27th Jan 21

Editors: Celine Carret / Zeljko Durdevic

Transaction Report:

19th Dec 2019

Dear Dr. Sinnis,

Thank you for the submission of your manuscript to EMBO Molecular Medicine. We have now heard back from the three referees whom we asked to evaluate your manuscript.

You will see below that the three referees find the topic of interest while sharing similar and overlapping concerns about the study. Ref. #2 particularly (and Ref. #1 also to some extent) would like to see additional experiments to increase the translational potentials and clinical relevance of the findings. Ref. #3 would appreciate additional read-outs and make interesting suggestions, needed to further improve the conclusiveness of the data. In addition, more controls, analyses and discussions/explanations are requested here and there that we agree would strengthen the manuscript.

We would therefore welcome the submission of a revised version within three months for further consideration and would like to encourage you to address all the criticisms raised as suggested to improve conclusiveness and clarity. Please note that EMBO Molecular Medicine strongly supports a single round of revision and that, as acceptance or rejection of the manuscript will depend on another round of review, your responses should be as complete as possible.

I look forward to receiving your revised manuscript.

Wishing you a merry Christmas and a happy and successful New Year, and all the best for the revision!

Yours sincerely,

Celine Carret

Celine Carret, PhD
Senior Editor
EMBO Molecular Medicine

***** Reviewer's comments *****

Referee #1 (Remarks for Author):

Hopp and collaborators analyze and characterize *P. falciparum* sporozoite motility in mouse skin and human skin xenografts, comparing it side-by-side to rodent malaria species. The authors propose that such an in vivo platform should be integrated in a system to validate vaccine candidates. This is quite an interesting manuscript that needs to be further strengthened to make sure that the authors' claims are fully supported by data.

I would have greatly appreciated to see some additional datasets, such as:

1. Vessel diameter in every frame, just to be sure that the differences are not confounded by presence of extra large or extra small vessels in some time point/experiments vs others
2. Study if the sporozoite delivered by mosquito bite behave similarly, since all experiments involve injected sporozoites. Just to address the fact that mosquitoes as they secrete saliva and anti-coagulants may evoke a mild inflammatory response in the dermis and test whether that could alter the nature of motility as they see with injected sporozoites.
3. Any information on immune response at the dermis, to address the 2A10 experiment results and also to address the difference in sporozoite release from the dermis in different *Plasmodium* species should be properly controlled.

Specific points

Figure 1 Legend - The CD31 to detect vessels is conjugated with Alexa fluor 647, but color in the image panel is magenta, which is a more UV range color. This is confusing. Why not use red in the panel or state the exact color in the legend?

Figure 2 Legend - The difference between tracks counted manually and automatically are too different in numbers. This looks like there is either a bias of exclusion in the manual analysis or a bias for oversampling in the automatic analysis. Both are problematic. Please justify the difference in numbers.

Figure 4 - Plotting speed Vs time is a measure of acceleration or deceleration. Under the standards of mathematical or physical representations of plots of this type, the units on both axes must be comparable. This means, measure time in a single unit, either min or sec. Mismatch of time units is confusing since, here the deceleration appears to be really quick, which is incorrect since speed is being measured with respect to time in sec, while the experiments is being followed in min.

"Reduced motility of *P. falciparum* sporozoites in the skin of passively immunized mice" section - Isotype control for 2A10 mAb should be used.

Referee #2 (Comments on Novelty/Model System for Author):

The technical quality is excellent. The authors have relied on their own already peer reviewed published data as a starting point for this current study. They have created new models and techniques to analyze sporozoite motility in the skin after intradermal injection. They have also created new parasite lines to assess sporozoite motility and these in conjunction with their modeling have enabled an exquisite understanding of the skin stage of the malaria life cycle.

The improved techniques are novel and build substantially upon the authors previous work.

The results strongly suggest that the mouse model of the skin stage of the malaria life cycle can be used to effectively determine the role that novel monoclonal or even polyclonal antibodies have on sporozoite exit from the skin. The authors show that a monoclonal antibody that targets the *Plasmodium falciparum* sporozoite and can effectively prevent motility, speed and final displacement. Thus, the model can be used to test the efficacy of novel antibodies to target the sporozoite and prevent its entry into the liver and ultimately prevent disease. However, it is currently unclear how good the model is as so little testing has been undertaken hence my "unclear at this stage" answer for #4. This is a big drawback.

Since malaria continues to sicken over 200 million people per annum and kill upwards of 450,000, this model will help move candidates forward for further testing and is relatively inexpensive.

Referee #2 (Remarks for Author):

The paper by Hopp and colleagues is a tour de force in understanding the short life of the *Plasmodium* sporozoite once it enters the skin after inoculation. The paper shows in great detail sporozoite motility, speed, displacement and invasion capability over time. The authors examine two rodent malaria parasites, *P. berghei* and *P. yoelii* as well as the human malaria parasite *P. falciparum*. Importantly, the authors show that the *P. falciparum* sporozoite behaves in a very similar manner in both mouse skin and xenografted human skin. In addition, the behavior of the *P. falciparum* sporozoite is similar to its rodent malaria counterparts suggesting that the skin stage of infection is not host specific. This thus allows for the testing of interventions on *P. falciparum* sporozoite activity in the skin in the mouse. This could potentially save significant amounts of money in pre-clinical testing as it would not require the use of expensive human-chimeric mice.

However, the title of the paper suggests that their model can be used for testing of interventions specifically for *P. falciparum* sporozoite activity and yet, they show only one experiment to address this hypothesis. Indeed, the authors ultimately conclude that their model would be ideal for the testing of antibody activity against the *P. falciparum* sporozoite in the mouse and the title also suggest this. To do this, they show that injection of mice with the 2A10 monoclonal antibody that recognizes the repeat region of the circumsporozoite protein leads to a significant decrease in sporozoite motility, speed and final displacement.

However, the authors do not show if these decreases ultimately prevent or reduce sporozoite

invasion of either lymphatic vessels or blood vessels. This information is an important addition to the paper and will help address the usefulness of the model. Additionally, antibody titration would also be extremely useful. I assume that decreasing the amount of antibody used for injection could decrease the effects on sporozoite activity and increasing the amount of antibody could possibly further increase the affects on sporozoite activity. These parameters need to be addressed.

In addition, I think it is essential that other antibodies are used in their assay other than 2A10. This could include further antibodies that target CSP as well as antibodies to sporozoite proteins, other than CSP to determine if these antibodies are also able to prevent sporozoite movement. For instance, do antibodies to TRAP (alone or in combination with antibodies to CSP) have an effect on the sporozoite skin stage phenotype. Other suggestions would be CeITOS, GAMA, P52 and P36.

Also, the authors speculate that small molecule inhibitors could also prevent sporozoite exit from the skin but do not show an example of this in their analysis. Presumably, this is possible.

In addition, since the authors tout the use of this model for testing of interventions, can they please add an additional figure which walks the reader through the process, showing the time necessary to gather, interpret and formalize the data analysis. How long would it take and how much would it cost to test a panel of 10 monoclonal antibodies, for instance, using their model.

I also have some more minor comments that I would like addressed.

1. Fig 1. Pb/Py, and just out of interest, what are the large green structures in the images?

2. Automated sporozoite detection and tracking. Figure 2 C&D

"Nonetheless, some differences in these two data sets are expected, given that the manually-tracked data set included only a subset of the total motile sporozoites that were tracked in the automated analysis."

"60 min (2 videos/77 manual tracks/23 automated tracks), and 120 min (2 videos/44 manual tracks/11 automated tracks)."

Could the authors please comment on why there are considerably fewer automated tracks generated at the 2 longer time courses with the automated tracking, given that all tracks are automatically recorded but only a subset are manually tracked. The low number of automatically generated tracks for these timepoints make statistical analysis of the data challenging, I assume.

3. Could the authors please explain what happens to an automated track when sporozoites overlap?

4. The automated and indeed manual data rely on clear visualization of sporozoites. Comparisons of motility are made across different species of sporozoites. It would be valuable to confirm that the fluorescence intensity of the sporozoites is similar. Different promoters and species are being used and therefore is it possible that certain differences in tracking results are due to differences in detection thresholds for sporozoites?

5. Center of origin Figure 3A-C

Could the authors please confirm that each center of origin plot shows the same number of tracks and what this number is. Do the number need to be consistent for these visual plots to be meaningfully interpreted? Also, what is the selection criteria for inclusion in the PoO plot? The first

n tracks from a selection of videos, etc?

6. "Videos of time courses were manually counted to determine the number of sporozoites that are motile and non-motile." Why were videos assessed manually when the preceding work has validated the use of the automated analysis and tracks all sporozoites, thereby providing this yes/no information based on track length? Or am I missing something?

Referee #3 (Comments on Novelty/Model System for Author):

Technical quality is as good as gets. No lab has compared three malaria species at the onset of an infection yet. Novelty is however somewhat compromised by recent work of the authors on *P. berghei* skin stage in mBio (PMID: 30459199) and others for developing similar tracking and analysis software. Especially the mBio paper reports on a *P. berghei* parasite expressing *P. falciparum* CSP. Without this paper I would be much more enthusiastic about the current one being published in EMBO Mol Med. Possible medical impact is still high as it immediately has consequences for malaria vaccine testing especially for non CSP targets.

Referee #3 (Remarks for Author):

In this well conducted and important study the authors investigate a large number of sporozoites from three different species of the malaria-causing parasites including the most important human-infecting *Plasmodium falciparum*. This is the first study describing human-infecting *P. falciparum* sporozoites in vivo. This in itself is of huge interest considering that the major surface protein (CSP) of the sporozoite is the protein constituting the first licensed malaria vaccine.

The paper compares in great detail how sporozoites of three different *Plasmodium* species migrate in the skin. It very importantly shows the egress (exit) rate from the skin by entering blood vessels. Remarkably this rate is the same in all three species. This is a very important finding for two reasons: first it shows that human-infecting malaria parasites can recognize and enter mice blood vessels. This makes it for me a valuable model to investigate *P. falciparum* sporozoites in the skin. Secondly, it also shows that the differences the authors could measure in the migration pattern and speed between the species is of little relevance. This, however, is not a critique, rather it is a very important message: the motility machinery of the parasite is apparently robust enough to be able to accommodate some level of change without impact on skin passage. I think it is important to clearly state this in the discussion, maybe even in the abstract, and use 'exit from the skin by blood vessel invasion' as the most important readout of the platform. It would be indeed interesting how reduction of migration correlates with blood vessel entry and whether the administration of anti-CSP antibodies act mainly through reducing migration or through killing parasites as recently suggested (Aliprandini 2018). Hence, my only major wish for this manuscript is to include data on sporozoite killing and blood vessel entry in the last experiments using an anti-CSP antibody as this is the key variable for an intervention study, as the authors have recently shown in a nice paper for *P. berghei* monoclonal anti CSP antibody 3D11 (mBio 2018). The authors can probably get this number from looking at the already available movies.

I also think the paper focuses too much on the platform biotechnology and hence buries very interesting scientific observations: Why not include the main result 'the skin is not a species specific barrier' in the title? This is an important result, as entry in the liver appears to require different

proteins.

Expand Figure 1 with some example time lapses showing migration in tissue and interaction with blood vessels

In videos 1-3 it would help the viewer if the authors annotate entry into lymphatic and blood vessels e.g. by distinct symbols?

Expand Figure 2 to include some of the processing steps described in Figure S2.

Please cite the ICY and FIJI papers.

State somewhere how much time you save per analyzed movie compared with manual tracking (rough estimate)

Figure 3 and corresponding text and elsewhere: I would suggest writing 10% instead of 9.8% and 8% instead of 7.7% etc, as the data does not have the precision to show digits past the comma.

Figure 5: maybe include a panel D that shows the relative decrease of the motility of the three species between 5 and 120 minutes, this makes the difference more readily apparent.

Figure 6: instead of maximum projections please choose a different way to portray blood vessel and lymph vessel invasion

Any chance to compare the time it takes for the sporozoites of the three species from the time of engagement with a blood vessel to blood vessel entry?

As stated above, Figure 8 should include data on blood vessel entry as this is the key outcome of skin migration. If this is not decreased a small decrease in motility is probably irrelevant and that would contradict the major message of the paper. I think this is the most important data that is lacking. This should also be discussed in context with published work showing decreased motility in vitro not affecting transmission.

Did the authors observe sporozoites dying similar to Aliprandini et al in the presence of the antibody? If yes, this should be shown in Figure 8 and mentioned in the text. If not, it should be discussed.

The larger decrease in % motile sporozoites compared with decrease in sporozoite speed was also observed on several mutants in vitro, which in some cases did not translate to a decrease of infectivity. This should be commented on in the discussion.

Discussion:

Please expanded to put previous work in the context of the current work.
e.g.:

A recent paper reveals an interaction between TRAP and an integrin (Dundas PNAS 2018) in *P. falciparum*. An interaction that is not observed in rodent parasites. Could this yield the differences in motile behavior? Is this interaction of relevance considering the similar blood vessel invasion rates between *P. berghei* and *P. falciparum* sporozoites?

Could this platform also be used to screen for or investigate drugs identified in vitro, e.g. Douglas et al Malaria J 2018

How does the paper compare to recent work by the Rustenberg laboratory?

References 24 and 35 miss the journal name

We thank all three reviewers for their enthusiastic support of our work and their helpful critiques. We have now performed additional experiments to address many of the reviewers' comments.

Our responses are in maroon below and additions or edits to the main manuscript are in red.

Referee #1 (Remarks for Author):

Hopp and collaborators analyze and characterize *P. falciparum* sporozoite motility in mouse skin and human skin xenografts, comparing it side-by-side to rodent malaria species. The authors propose that such an in vivo platform should be integrated in a system to validate vaccine candidates. This is quite an interesting manuscript that needs to be further strengthened to make sure that the authors' claims are fully supported by data.

I would have greatly appreciated to see some additional datasets, such as:

1. Vessel diameter in every frame, just to be sure that the differences are not confounded by presence of extra large or extra small vessels in some time point/experiments vs others

Though the movies in which we demonstrate blood vessel entry for our readers are performed by staining the blood vessels prior to the experiment, having two fluorescent channels significantly slows down acquisition time on the confocal microscope, and the majority of our movies are made without staining the vessels. In fact, due to higher acquisition speed, it is much easier to score vessel invasion in movies in which the vessels are not stained. As a result, we cannot go back and generate vessel diameter data from the majority of the movies we used for this study. However, we do not believe that our data are confounded by extra large or extra small vessels (with one exception outlined below) for the following reasons:

1. Dermal capillaries are fairly consistent in size, between 5-7.5 microns in internal diameter. Additionally, the skin circulatory system has a consistent architecture, which has been described in detail (Braverman *et al.*, *J Invest Dermatol* 93; 1989 and Braverman *et al.*, *J Invest Dermatol Symposium Proceedings* 5; 2000).

2. We have scored blood vessel entry events in a large number of movies for each *Plasmodium* species (22 to 46 movies), so that any variation in the size of the vessels in a particular movie should be averaged into the mix and not skew the data.

We would, however, like to mention our observations in the grafted human skin. We found that *Pf* sporozoites moved significantly less distance and entered blood vessels at a greater frequency in grafted human skin compared to mouse skin. We hypothesized that this was due to the abnormally large blood vessels in the grafts due to anastomoses between invading mouse blood vessels and the resident human blood vessels. Indeed, the wider and abnormally shaped vessels in the graft can be appreciated by comparing videos S9 - S11 with all of the other videos. Furthermore, somewhat abnormal sporozoite entry can be observed in video S11 in which vessel entry by *Pf* sporozoites in the human skin graft are shown. This is all to say that our intensive work with these videos has enabled us to pick up on these types of issues and in mouse skin we never observed anything similar to this. We agree with the reviewer that this deserves more attention in the manuscript and have now added the following text:

As shown in Figure 7F, more than 10% of motile *P. falciparum* sporozoites enter blood vessels in the human skin graft compared with 5.5% of motile *P. falciparum* in mouse skin. Note that there was no lymphatic invasion in the skin graft since lymphatic vessels are not reconstituted after surgery (25). This higher level of blood vessel entry may be due to the somewhat abnormal anatomy of the blood vessels in the grafted skin. Indeed, comparison of the blood vessels in the engrafted skin (Videos S9-S11) with those of normal mouse skin (Videos S1-S8), show areas of widening and irregular shaped vessels in the graft.

Thus, it is likely the anatomy of the grafted and anastomosed vessels in the human skin graft that is responsible for the decreased displacement and speed of sporozoites in this setting. Though the human skin xenograft model is being increasingly used in skin physiology and disease studies (reviewed in Refs 33, 34), the nature of the anastomosing blood vessels remains poorly understood.

2. Study if the sporozoite delivered by mosquito bite behave similarly, since all experiments involve injected sporozoites. Just to address the fact that mosquitoes as they secrete saliva and anti-coagulants may evoke a mild inflammatory response in the dermis and test whether that could alter the nature of motility as they see with injected sporozoites.

We agree with the Reviewer that this is an important question. We have addressed this in our previous study and found no dramatic differences in the pattern of motility when sporozoites are deposited by mosquito bite or needle (Hopp et al., eLife 2015; Fig 1- Fig Supplement 2). While additional experimentation is needed to look for the more subtle effects of mosquito saliva, it is difficult to acquire the large amounts of data necessary to tease apart more subtle effects because of technical issues in capturing sporozoites as they are inoculated by a probing mosquito. While we have performed a comparison of sporozoite motility after inoculation by needle versus mosquito bite for the rodent parasites, we agree that it would be nice to do this with *P. falciparum*. Unfortunately, these experiments cannot be performed in any of the confocal facilities on campus due to biosafety issues.

3. Any information on immune response at the dermis, to address the 2A10 experiment results and also to address the difference in sporozoite release from the dermis in different Plasmodium species should be properly controlled.

We agree that this is indeed both interesting and important. Indeed, it is an area of current investigation in our laboratory. Only one prior study has addressed the dermal immune response after sporozoite inoculation into the skin by needle injection (Mac-Daniel L et al., J Immunol 2014) and observed an inflammatory infiltration – with neutrophils beginning to arrive at 2 hrs post inoculation and plateauing at 4 hrs and monocytes arriving significantly later, with increased numbers at 24 hrs. There are no studies on the impact of the dermal innate response on sporozoite infectivity and this is an area we are currently investigating. Thus far, we have found that depletion of neutrophils with the 1A8 antibody, and depletion of mast cells (using Kit^{w/w-v} mice), do not impact sporozoite infectivity after mosquito inoculation (see graphs below). Thus, we hypothesize that the majority of sporozoites exit prior to the entry of neutrophils and monocytes, thus escaping destruction by the innate response. While we agree that further investigations into this area are of significant interest, this was not within the scope of this present study.

P. berghei liver infection following infected mosquito bite

Specific points

Figure 1 Legend - The CD31 to detect vessels is conjugated with Alexa fluor 647, but color in the image panel is magenta, which is a more UV range color. This is confusing. Why not use red in the panel or state the exact color in the legend?

We completely agree that this is a valid point. We have changed the labelling from “far red” to “magenta” in all instances used. CD31 was indeed detected with an Alexa fluor 647-conjugated antibody and sporozoites expressed red fluorophores (mCherry and TdTomato). Since dark red/bright red was not a feasible color combination, we chose to use the present color combination, which is one that is encouraged to use by many journals, as it accommodates color-blind readers.

Figure 2 Legend - The difference between tracks counted manually and automatically are too different in numbers. This looks like there is either a bias of exclusion in the manual analysis or a bias for oversampling in the automatic analysis. Both are problematic. Please justify the difference in numbers.

As described in the results section (page 4&5), a key difference between manually tracked and automatically acquired sporozoite tracks, is that only a subset of the total motile sporozoites were acquired manually. While potentially problematic, we have addressed these issues by a thorough side-by-side comparison and some additional analyses shown in Fig S3:

“Nonetheless, some differences in these two data sets are expected, given that the manually-tracked data set included only a subset of the total motile sporozoites that were tracked in the automated analysis: While the automated analysis tracks all motile sporozoites, therefore generating tracks of different total durations, the manual tracking analysis only included motile sporozoites that were observed in the field of view for the full duration of the 4 min long video, which was a necessary requirement to allow calculation of mean square displacement (3). To be able to directly compare sporozoite displacement for tracks of different durations, displacement was normalized to the average displacement per 30-second interval, thus allowing comparison of the automated and manual tracking data sets. As previously described (3), sporozoite displacement drops

at 20 min after inoculation in both data sets (Figure 2E-F). However, automated tracking resulted in statistically higher displacements at the latest time point, 120 minutes. This is likely due to inclusion of all motile sporozoites in the automated tracking data set, because sporozoites that leave the field of view would be predicted to have higher displacements than those that stay in the field. To determine if this was the case, sporozoites that were leaving the field of view and thus excluded from the manual analysis, were manually tracked for the 10 min and 120 min time points and added to the previous manual analysis. This showed that the displacement of the total sporozoite population at both 10 min and 120 min after inoculation is higher than was suggested by the original analysis of sporozoites that remain in the field of view throughout the acquired video (Figure S3).”

Figure 4 - Plotting speed Vs time is a measure of acceleration or deceleration. Under the standards of mathematical or physical representations of plots of this type, the units on both axes must be comparable. This means, measure time in a single unit, either min or sec. Mismatch of time units is confusing since, here the deceleration appears to be really quick, which is incorrect since speed is being measured with respect to time in sec, while the experiments is being followed in min.

Acceleration is calculated when a change in speed is divided by a time interval ($a = \Delta\text{speed} \div \Delta\text{time}$). However, what is plotted in Figure 4 is the sporozoite speed, calculated as the total track length, divided by the track duration. This was done at several time points and the time interval between these time points was not used for the calculation. We hope this clarifies the point that what is graphed is in fact not acceleration.

"Reduced motility of *P. falciparum* sporozoites in the skin of passively immunized mice" section - Isotype control for 2A10 mAb should be used.

We agree and have now performed additional experiments with an isotype control, VRC01, an anti-HIV antibody of the same isotype as the additional mAbs we have now tested. Importantly, we saw no differences between this isotype control and no antibody, which was used as a control in our initial mAb 2A10 experiments. These data are now included in the new Figure 8.

Referee #2 (Comments on Novelty/Model System for Author):

The technical quality is excellent. The authors have relied on their own already peer reviewed published data as a starting point for this current study. They have created new models and techniques to analyze sporozoite motility in the skin after intradermal injection. They have also created new parasite lines to assess sporozoite motility and these in conjunction with their modeling have enabled an exquisite understanding of the skin stage of the malaria life cycle.

The improved techniques are novel and build substantially upon the authors previous work.

The results strongly suggest that the mouse model of the skin stage of the malaria life cycle can be used to effectively determine the role that novel monoclonal or even polyclonal antibodies have on sporozoite exit from the skin. The authors show that a monoclonal antibody that targets the *Plasmodium falciparum* sporozoite and can effectively prevent motility, speed and final displacement. Thus, the model can be used to test the efficacy of novel antibodies to target the sporozoite and prevent its entry into the liver and ultimately prevent disease. However, it is

currently unclear how good the model is as so little testing has been undertaken hence my "unclear at this stage" answer for #4. This is a big drawback.

We thank the reviewer for their appreciation of our hard work! We also agree that an additional antibody would make this a stronger study and have now added a control antibody and two other CSP antibodies given to us by Bob Seder. We have also extended our analyses to include blood vessel entry, which we agree is likely the most important readout for inhibitory activity. These data are now included in the revised Figure 8.

Since malaria continues to sicken over 200 million people per annum and kill upwards of 450,000, this model will help move candidates forward for further testing and is relatively inexpensive.

We agree!

Referee #2 (Remarks for Author):

The paper by Hopp and colleagues is a tour de force in understanding the short life of the Plasmodium sporozoite once it enters the skin after inoculation. The paper shows in great detail sporozoite motility, speed, displacement and invasion capability over time. The authors examine two rodent malaria parasites, *P. berghei* and *P. yoelii* as well as the human malaria parasite *P. falciparum*. Importantly, the authors show that the *P. falciparum* sporozoite behaves in a very similar manner in both mouse skin and xenografted human skin. In addition, the behavior of the *P. falciparum* sporozoite is similar to its rodent malaria counterparts suggesting that the skin stage of infection is not host specific. This thus allows for the testing of interventions on *P. falciparum* sporozoite activity in the skin in the mouse. This could potentially save significant amounts of money in pre-clinical testing as it would not require the use of expensive human-chimeric mice.

However, the title of the paper suggests that their model can be used for testing of interventions specifically for *P. falciparum* sporozoite activity and yet, they show only one experiment to address this hypothesis. Indeed, the authors ultimately conclude that their model would be ideal for the testing of antibody activity against the *P. falciparum* sporozoite in the mouse and the title also suggest this. To do this, they show that injection of mice with the 2A10 monoclonal antibody that recognizes the repeat region of the circumsporozoite protein leads to a significant decrease in sporozoite motility, speed and final displacement.

However, the authors do not show if these decreases ultimately prevent or reduce sporozoite invasion of either lymphatic vessels or blood vessels. This information is an important addition to the paper and will help address the usefulness of the model. Additionally, antibody titration would also be extremely useful. I assume that decreasing the amount of antibody used for injection could decrease the effects on sporozoite activity and increasing the amount of antibody could possibly further increase the affects on sporozoite activity. These parameters need to be addressed.

We completely agree with the reviewer and have now counted our original videos and added blood vessel entry data for 2A10 (150 micrograms) compared to naïve mice. This is now included in the revised Figure 8. While we did perform two experiments with 300 micrograms of 2A10 and saw little to no motility in the 2A10 inoculated mice, the videos were of poor quality and we would prefer not to include them. With the COVID pandemic the IRB for human blood donors, that we require to grow mosquito-infectious gametocytes was put on hold and purchased blood, being less fresh, has

not been a good substitute. Thus, our *P. falciparum* cycles have not been as robust and we have chosen to prioritize the experiments with the additional antibodies (see below).

In addition, I think it is essential that other antibodies are used in their assay other than 2A10. This could include further antibodies that target CSP as well as antibodies to sporozoite proteins, other than CSP to determine if these antibodies are also able to prevent sporozoite movement. For instance, do antibodies to TRAP (alone or in combination with antibodies to CSP) have an effect on the sporozoite skin stage phenotype. Other suggestions would be CelTOS, GAMA, P52 and P36.

Surprisingly there are no other commercially or publicly available CSP-specific antibodies, which is why we focused on mAb 2A10. However, we completely agree with the reviewer and therefore asked Dr. Bob Seder, VRC/NIH for two of the antibodies he recently described (Kisalu *et al.*, *Nat Med*, 2018: PMID: 29554083). We have now performed intravital imaging with these antibodies, CIS43 and mAb10, and *P. falciparum* sporozoites and these data are now included in a revised Figure 8.

While we agree that it would be nice to test antibodies specific for other sporozoite proteins. We could not find any inhibitory antibodies specific for the *P. falciparum* proteins suggested by the reviewer with the exception of CelTOS. The CelTOS antibodies are currently not being shared with the community. This would be an interesting avenue of investigation for a follow up study and would require us to generate inhibitory antibodies to these proteins, which we feel is beyond the scope of this study.

Also, the authors speculate that small molecule inhibitors could also prevent sporozoite exit from the skin but do not show an example of this in their analysis. Presumably, this is possible.

We agree and have used our intravital imaging platform to test a motility inhibitor we found in a Pathogen Box screen against *P. falciparum* sporozoite motility in mouse skin. Some of these data are shown below, however, we would prefer not to include them in this paper as we were planning to include them in a paper currently being written that describes our high throughput in vitro motility screen.

In addition, since the authors tout the use of this model for testing of interventions, can they please add an additional figure which walks the reader through the process, showing the time necessary to gather, interpret and formalize the data analysis. How long would it take and how much would it cost to test a panel of 10 monoclonal antibodies, for instance, using their model.

This is a very valuable point and we have now included a new Expanded View Table outlining each step with the time and cost required.

I also have some more minor comments that I would like addressed.

1. Fig 1. Pb/Py, and just out of interest, what are the large green structures in the images?

These are hair follicles which have a strong autofluorescence in the green channel. We have now added this to the legend of Figure 1:

Maximum intensity projection over 240 sec shows trajectories of moving sporozoites (green) and blood vessels (magenta). The light green structures in the first two panels are autofluorescent hair follicles. Scale bar, 50 μ m. See videos S1-S3.

2. Automated sporozoite detection and tracking. Figure 2 C&D

"Nonetheless, some differences in these two data sets are expected, given that the manually-tracked data set included only a subset of the total motile sporozoites that were tracked in the automated analysis."

"60 min (2 videos/77 manual tracks/23 automated tracks), and 120 min (2 videos/44 manual tracks/11 automated tracks)."

Could the authors please comment on why there are considerably fewer automated tracks generated at the 2 longer time courses with the automated tracking, given that all tracks are automatically recorded but only a subset are manually tracked. The low number of automatically generated tracks for these timepoints make statistical analysis of the data challenging, I assume.

We thank Reviewer 2 for pointing this out! We mistakenly included the wrong graph for the automated speed tracking, using a graph that we had made for another purpose that did not include the data from all of the videos used for the automated displacement graph. We have now changed the automated speed graph to include all data and have modified the figure legend accordingly:

C: 5 min (3 videos/37 manual tracks/203 automated tracks), 10 min (6 videos/112 manual tracks/292 automated tracks), 20 min (4 videos/95 manual tracks/184 automated tracks), 30 min (6 videos/63 manual tracks/235 automated tracks), 60 min (7 videos/77 manual tracks/162 automated tracks), and 120 min (6 videos/44 manual tracks/151 automated tracks).

3. Could the authors please explain what happens to an automated track when sporozoites overlap?

This is a good question as the ability of ICY to accurately track sporozoites, including overlapping sporozoites, was the one dramatic advantage of this software. Other imaging software that we tried, Image J and Imaris, were not able to do this. We do not know how exactly ICY can keep track

of sporozoites, perhaps it factors in directional vectors? Nonetheless, the displacement data in Figure 2C suggests that overlapping tracks are not an issue.

4. The automated and indeed manual data rely on clear visualization of sporozoites. Comparisons of motility are made across different species of sporozoites. It would be valuable to confirm that the fluorescence intensity of the sporozoites is similar. Different promoters and species are being used and therefore is it possible that certain differences in tracking results are due to differences in detection thresholds for sporozoites?

The need for strong sporozoite fluorescence to accurately track sporozoites was what prompted us to generate the Tdtomato Pf sporozoites, an undertaking that took several years (McLean et al., Sci Rep; PMID: 31511546). Though we have not directly compared the fluorescence intensity of the Pb, Py, and Pf transgenic parasites, we used the same exposure times and laser intensities for all of our experiments indicating that they are similar. Indeed, we have found that the variations in fluorescence intensity due to the different depths at which the sporozoites are located, create more variation than differences in fluorescence intensity among the different species of sporozoites.

5. Center of origin Figure 3A-C

Could the authors please confirm that each center of origin plot shows the same number of tracks and what this number is. Do the number need to be consistent for these visual plots to be meaningfully interpreted? Also, what is the selection criteria for inclusion in the PoO plot? The first n tracks from a selection of videos, etc?

The center of origin plots shown in Figure 3A-C show the same number of tracks per timepoint for all species. 5 min: 40 tracks; 10 min: 32 tracks; 30 min: 25 tracks; 60 min: 21 tracks. These tracks were taken from time course experiments where 4 min videos were performed on the same population of sporozoites at each of the indicated time points. As expected, track number decreases over time. Since the Pb video had fewer tracks than the Py and Pf videos, tracks were randomly deleted (using the ICY software), making sure that the displacement displayed in each star plot matched the displacement of the entire sporozoite population at that time point.

We have now added some explanatory text to the Figure 3 legend:

Tracks generated through automated tracking were plotted to a common origin, to visualize parasite dispersal *P. berghei* (A), *P. yoelii* (B) and *P. falciparum* (C) at 5, 10, 30 and 60 min after intradermal inoculation. For each timepoint, the tracks are representative of the average displacement of the entire sporozoite population. Track numbers used for each timepoint were: 5 min, 40 tracks; 10 min, 32 tracks; 30 min, 25 tracks; 60 min, 21 tracks.

6. "Videos of time courses were manually counted to determine the number of sporozoites that are motile and non-motile." Why were videos assessed manually when the preceding work has validated the use of the automated analysis and tracks all sporozoites, thereby providing this yes/no information based on track length? Or am I missing something?

For the automated tracking, sporozoite movement was the key parameter by which we got the software to accurately identify sporozoites. Thus, this analysis does not include non-motile sporozoites and so percent motile must be done manually. To a human, a non-motile sporozoite is pretty clear and can easily be differentiated from fluorescent background objects, by size and shape. This was not possible to do with the software (we tried!).

Referee #3 (Comments on Novelty/Model System for Author):

Technical quality is as good as gets. No lab has compared three malaria species at the onset of an infection yet. Novelty is however somewhat compromised by recent work of the authors on *P. berghei* skin stage in mBio (PMID: 30459199) and others for developing similar tracking and analysis software. Especially the mBio paper reports on a *P. berghei* parasite expressing *P. falciparum* CSP. Without this paper I would be much more enthusiastic about the current one being published in EMBO Mol Med. Possible medical impact is still high as it immediately has consequences for malaria vaccine testing especially for non CSP targets.

While the mBio paper was an important proof of concept on the role of antibody at the dermal inoculation site, it used the rodent malaria model. Though *P. berghei*-expressing PfCSP is a valuable tool, one would not make decisions about moving forward with a vaccine or drug unless it was tested on human malaria sporozoites. Indeed, we have recently screened the pathogen box compounds for their impact on *P. falciparum* sporozoite motility and found compounds that inhibited Pf, but not Pb, and vice versa. Therefore, we do think that the development of an in vivo system to look at the impact of antibodies and drugs on *P. falciparum* sporozoites represents a real advance. Further, we agree with this reviewer's assessment that one of the most interesting aspects of this study is that the skin is not a species-specific barrier for the parasite and that while the platform development is important, it is not the only important discovery of this study. We address this further below.

It should be noted that the studies in the mBio paper were being performed in parallel with this study. Because another group was performing an overlapping study with *P. berghei*, we rushed to get the mBio paper out but made a condition of publication that we would not include the methods for automated tracking, which were developed specifically for the current study. Though the method we developed was used in that paper, the methodology was not included and is described for the first time in this manuscript, which unfortunately has been significantly delayed because of postdoc moves, the pandemic and the concomitant shut-down of our insectary and microscopy facilities.

Referee #3 (Remarks for Author):

In this well conducted and important study the authors investigate a large number of sporozoites from three different species of the malaria-causing parasites including the most important human-infecting *Plasmodium falciparum*. This is the first study describing human-infecting *P. falciparum* sporozoites in vivo. This in itself is of huge interest considering that the major surface protein (CSP) of the sporozoite is the protein constituting the first licensed malaria vaccine.

The paper compares in great detail how sporozoites of three different *Plasmodium* species migrate in the skin. It very importantly shows the egress (exit) rate from the skin by entering blood vessels. Remarkably this rate is the same in all three species. This is a very important finding for two reasons: first it shows that human-infecting malaria parasites can recognize and enter mice blood vessels. This makes it for me a valuable model to investigate *P. falciparum* sporozoites in the skin. Secondly, it also shows that the differences the authors could measure in the migration pattern and speed between the species is of little relevance. This, however, is not a critique, rather it is a very

important message: the motility machinery of the parasite is apparently robust enough to be able to accommodate some level of change without impact on skin passage. I think it is important to clearly state this in the discussion, maybe even in the abstract, and use 'exit from the skin by blood vessel invasion' as the most important readout of the platform.

We agree. We have now added text to both the abstract and discussion to reflect this and have changed the title of the manuscript.

Abstract:

The combined data suggest that in contrast to the liver and blood stages, the skin is not a species-specific barrier for *Plasmodium*. Indeed, *P. falciparum* sporozoites entered blood vessels in mouse skin at similar rates to the rodent malaria parasites. Furthermore, we demonstrate that antibodies targeting sporozoites significantly impact the motility of *P. falciparum* sporozoites in mouse skin.

Discussion on page 20:

These data suggest that sporozoite recognition of blood vessels is likely based on molecules that are shared among different host species, however, precisely what sporozoites are sensing is not known. It also indicates that the parasite's motility machinery is sufficiently robust to enable them to enter vessels of different species. Thus, we demonstrate that there are critical cellular and molecular commonalities between mice and primates during the skin phase of the life cycle.

It would be indeed interesting how reduction of migration correlates with blood vessel entry and whether the administration of anti-CSP antibodies act mainly through reducing migration or through killing parasites as recently suggested (Aliprandini 2018). Hence, my only major wish for this manuscript is to include data on sporozoite killing and blood vessel entry in the last experiments using an anti-CSP antibody as this is the key variable for an intervention study, as the authors have recently shown in a nice paper for *P. berghei* monoclonal anti CSP antibody 3D11 (mBio 2018). The authors can probably get this number from looking at the already available movies.

We agree that blood vessel entry by *P. falciparum* sporozoites in the presence of antibody is an important dataset to generate. We have now gone back to our movies and scored them for blood vessel entry in the presence and absence of antibody and those data are included in the revised Figure 8. Regarding the difference between sporozoite killing and reducing migration leading to reduced blood vessel entry: At the time points we assay, from 10 to 20 minutes after inoculation, we have never observed the broken up sporozoites or dotted fluorescence observed by Aliprandini *et al.*, *Nat Microbiology*, 2018. Thus, we score motile/non-motile as a proxy for an inability to enter vessels. Even if sporozoites are not actively killed, if they do not enter vessels, they will ultimately die in the skin.

I also think the paper focuses too much on the platform biotechnology and hence buries very interesting scientific observations: Why not include the main result 'the skin is not a species specific barrier' in the title? This is an important result, as entry in the liver appears to require different proteins.

We agree with the reviewer that the behavior of human malaria parasites in rodent skin is an important finding, perhaps equally important to the development of a testing platform. We have now changed the title of the paper:

Comparative intravital imaging of human and rodent malaria sporozoites reveals the skin is not a species-specific barrier

Expand Figure 1 with some example time lapses showing migration in tissue and interaction with blood vessels

Maximum projections of blood vessel interactions are shown in Figure 6 where we think they fit best since that is where we show the blood vessel entry data. We do not think that additional maximum projections would elucidate much re: tissue migration since these projections can get very busy. This is why we include so many supplemental movies in our manuscript which are the best illustrations of these behaviors.

In videos 1-3 it would help the viewer if the authors annotate entry into lymphatic and blood vessels e.g. by distinct symbols?

For any given movie there are very few, sometimes no blood vessel and lymphatic entry events. Thus, we did not modify videos 1-3 to include these symbols since these videos have very very few events and the point of these videos is to show migration through the tissue.

Expand Figure 2 to include some of the processing steps described in Figure S2.

We have now merged Supplemental Figure 2 with Figure 2 in the main manuscript and changed the legend, etc accordingly.

Please cite the ICY and FIJI papers.

We agree, these papers need to be cited and we now added references for both: de Chaumont *et al.*, *Nat Meth*, 2012 and Schindelin *et al.*, *Nat Meth*, 2012.

State somewhere how much time you save per analyzed movie compared with manual tracking (rough estimate)

Reviewer 2 had a similar question and we have now included a flow chart of the steps with time and cost as a new Expanded View Table.

Figure 3 and corresponding text and elsewhere: I would suggest writing 10% instead of 9.8% and 8% instead of 7.7% etc, as the data does not have the precision to show digits past the comma.

While the reviewer is correct, the percentages are calculated from a series of percentages and we have rounded them to one unit after the decimal point. Since the differences among species are small, we feel that these are helpful.

Figure 5: maybe include a panel D that shows the relative decrease of the motility of the three species between 5 and 120 minutes, this makes the difference more readily apparent.

We agree with the reviewer that the changes in motility over time are not easy to discern in the original figure and have now added a panel D with these data for the 60 and 120 min time points (the percent motile sporozoites remains relatively constant until 60 min post-inoculation). We have also changed the corresponding text in the manuscript on page 12:

We found that compared to *P. berghei*, which, over the course of two hours, lose the ability to move in the dermis, the percentage of motile *P. yoelii* and *P. falciparum* does not rapidly drop off and remains similar at later time points. ~~with 30% to 20% of sporozoites moving at 60 to 120 min after inoculation, respectively.~~

To better illustrate this, we calculated the fold-decrease in motile sporozoites at 60 and 120 min (Figure 5B): The percent of motile *P. berghei* sporozoites drops 3 and 4.5 fold at 60 and 120 min post-inoculation, respectively. In contrast, over the same time frame, the number of motile *P. yoellii* and *P. falciparum* sporozoites decreases 1.4 fold and ~2 fold.

Figure 6: instead of maximum projections please choose a different way to portray blood vessel and lymph vessel invasion

The maximum projections in Figure 6 were meant to illustrate sporozoite behavior around the vessels and we have now changed the text to clarify (see addition below). The only way to show vessel entry is with the movies that were included as supplemental files (Videos S4-S6). The text on page 13 now reads:

We found that similar to *P. berghei*, both *P. yoellii* and *P. falciparum* engage with CD31 blood vessels by frequently circling around the vessel (Videos S4-S6). This is illustrated in the maximum projections of sporozoites moving in the vicinity of blood vessels in Figure 6A.

Any chance to compare the time it takes for the sporozoites of the three species from the time of engagement with a blood vessel to blood vessel entry?

We agree that this would be very interesting. However, the majority of our blood vessel entry events were scored using movies in which blood vessels were not stained, making these kinds of studies not possible given the limited numbers of movies with CD31-stained vessels. The reason for this is that when we have an additional channel to acquire on the confocal scope, it significantly slows down acquisition time and actually makes blood vessel entry events more difficult to score.

As stated above, Figure 8 should include data on blood vessel entry as this is the key outcome of skin migration. If this is not decreased a small decrease in motility is probably irrelevant and that would contradict the major message of the paper. I think this is the most important data that is lacking. This should also be discussed in context with published work showing decreased motility in vitro not affecting transmission.

We completely agree and we have now included blood vessel entry data in the revised Figure 8. We are not aware of work showing that decreased motility in vitro does not affect transmission, when transmission is quantified. Transmission studies have to be interpreted with caution: Many investigators perform these studies in a binary fashion – can a blood stage infection be initiated by a large number of mosquito bites or the inoculation of a large number of dissected sporozoites initiate a blood stage infection after IV inoculation? Sporozoite mutants that are fairly attenuated and would likely only last 2 or 3 generations, can initiate a blood stage infection when inoculated IV. Its critical, to properly evaluate a motility defect quantitatively, testing a range of sporozoite doses, particularly low doses since infection is generally initiated by one mosquito bite or a low number of sporozoites. If we have missed some literature, it would be important for the reviewer to let us know and we'd be happy to address it.

Did the authors observe sporozoites dying similar to Aliprandini et al in the presence of the antibody? If yes, this should be shown in Figure 8 and mentioned in the text. If not, it should be discussed.

We have never observed the broken up sporozoites observed by Aliprandini et al. While this may be due to the particular antibody they used, it is likely because of differences in our experimental

design. We inject antibody one day prior to sporozoite inoculation and visualize sporozoites 10 min after they are inoculated. In contrast, the Aliprandini paper which was much more focused on the immune response to sporozoites in the skin, followed sporozoites in antibody-inoculated mice for over 30 minutes. While it would be very interesting for us to extend our experiments in this way, particularly for *P. falciparum* sporozoites, we feel that this is not within the scope of the current study and would require a bonafide study of its own.

The larger decrease in % motile sporozoites compared with decrease in sporozoite speed was also observed on several mutants *in vitro*, which in some cases did not translate to a decrease of infectivity. This should be commented on in the discussion.

It would be helpful to know what sporozoite mutants and what studies the reviewer is referring to. We are not aware of any mutants that demonstrate an in vivo motility defect and have normal infectivity when they are inoculated into the skin. The coronin mutant from the Frischknecht laboratory had *in vitro* motility defects but did not have *in vivo* motility defects, likely because its *in vitro* defect was in adhesion to the glass coverslip and *in vivo* this is compensated by the 3-dimensions of the extracellular matrix. Thus, we would not expect an *in vivo* infectivity defect given that its motility *in vivo* was not compromised. Nonetheless, they only tested for *in vivo* infectivity after 10 mosquito bites and it would be informative to also look at infectivity after 5 and 1 mosquito bites. If we are missing something, we would be happy for the reviewer to let us know and to modify our manuscript accordingly.

Discussion:

Please expanded to put previous work in the context of the current work.

We have now added a discussion of the Roestenberg paper and the Douglas paper to the last paragraph of the Discussion on page 22, which now reads:

This is a significant step forward and has several advantages over currently used assays for pre-clinical testing of inhibitory activity on *P. falciparum* sporozoites. *In vitro* assays quantifying cell traversal and invasion suffer from the low infectivity of *P. falciparum* sporozoites *in vitro* and our lack of knowledge on how to activate *P. falciparum* sporozoites for these processes once they have been dissected from mosquito salivary glands. A recent study visualized *P. falciparum* sporozoites in human skin explants and developed an automated tool to track their movements (41), yet the lack of blood supply to the tissue limits the informative observation period and does not allow for the assessment of blood vessel entry, a critical component of infectivity (41). Significant progress has been made with the development of a humanized mouse model (42), however, the mice used for these assays are expensive and immunocompromised, making it difficult to evaluate vaccine candidates. Our model, using immunocompetent mice, enables the use of larger numbers of mice due to their low-cost relative to the humanized mice, and evaluation of vaccine candidates due to their fully competent immune system. This model could also be used to evaluate compounds for their inhibitory effect on *P. falciparum* sporozoite motility *in vivo*. *In vitro* screens with the rodent parasite *P. berghei*, such as the one performed by Douglas *et al.* (43) identified several inhibitory compounds of motility, which could now be further tested on? *P. falciparum* sporozoites *in vivo*. Moreover, the automated sporozoite tracking tools described in this paper and the study by Winkel *et al* (41) facilitate

more rapid data analysis. Together this makes for a platform that will enable higher throughput testing of vaccines, chemical inhibitors, and antibodies targeting human malaria sporozoites, prior to costly primate or human studies.

A recent paper reveals an interaction between TRAP and an integrin (Dundas PNAS 2018) in *P. falciparum*. An interaction that is not observed in rodent parasites. Could this yield the differences in motile behavior? Is this interaction of relevance considering the similar blood vessel invasion rates between *P. berghei* and *P. falciparum* sporozoites?

While the biochemistry in the Dundas paper is outstanding, and clearly shows that PfTRAP interacts with alphaV beta3 integrins, there was no corresponding phenotype with either *P. berghei* parasites expressing PfTRAP, or with *P. falciparum* sporozoites. Our group performed these experiments and we tried numerous in vitro and in vivo assays in order to discern a phenotype related to alphaV beta3 integrin expression and PfTRAP. However, after a significant amount of work, we only observed that Pf sporozoites moved somewhat faster and further in integrin-knockout mice compared to controls. Even this was a small effect and we saw no other phenotype in the skin, including any change in blood vessel entry. Likely the faster movement was due to a looser extracellular matrix, though this was not explored. Thus, we do not believe this interaction is of relevance to blood vessel entry, which fits with the similar blood vessel entry rates we observe between *P. berghei* and *P. falciparum* in this study.

Could this platform also be used to screen for or investigate drugs identified in vitro, e.g. Douglas et al Malaria J 2018

Yes it can and we are beginning to do this with compounds that inhibit *P. falciparum* motility in vitro. See a graph of some preliminary results we included as a response to a similar question posed by Reviewer 2. We have now added this and the Douglas study to the Discussion (see above).

How does the paper compare to recent work by the Rustenberg laboratory?

We agree that this should have been included and have now added this to the Discussion (see above).

References 24 and 35 miss the journal name

Done

11th Jan 2021

Dear Dr. Sinnis,

Thank you for the submission of your revised manuscript to EMBO Molecular Medicine. I am pleased to inform you that we will be able to accept your manuscript pending the following final amendments:

- 1) Please address all the points raised by the referee #2.
- 2) Movies: Rename movie files to "Movie EV1" etc. (also in the text) and zipp their legends to the respective movie file.
- 3) Tables: Remove Table S2 legend from EV Table 1 file.
- 4) In the main manuscript file, please do the following:
 - Correct/answer the track changes suggested by our data editors by working from the attached/uploaded document.

***** Reviewer's comments *****

Referee #1 (Remarks for Author):

The manuscript contains very interesting findings and the authors made a great job in answering all my comments and suggestions.

Referee #2 (Comments on Novelty/Model System for Author):

The use of this as a platform for the testing of antibody interventions is partially demonstrated and the possible application to motility inhibitory compounds is clear.

Although in vivo imaging has been carried out before, this shows the first instance of automated tracking in human skin with a human infecting parasite.

The model can be used to assess the ability of antibodies and other interventions to reduce sporozoite motility in the skin, with the concomitant application to vaccine research. However, the human monoclonal antibodies used in the follow up study did not lead to a statistically significant drop in sporozoite motility or vessel entry which is of concern. It is possible that this is due to their clearance in the mouse after passive transfer and this should be analyzed. The limited number of animals used is also not ideal.

The manuscript is extremely well written and clear.

Referee #2 (Remarks for Author):

The authors have made their best effort to address the issues raised by this reviewer and also by the further reviewers. There are two minor comments I would like addressed.

1. Figure 5. The large decrease in sporozoites at the later timepoints for Pb is interesting. Pb is known to be a less host specific parasite than Py and Pf. CD81 for example is a hepatocyte receptor required by Py and Pf but not for Pb, and indeed Pb will infect non hepatocyte cell lines in vitro. Is it possible that the more rapid decrease in motility in the skin over time observed in Pb may be related to an early switch to inappropriate invasion, which is not observed in Py or Pf? Could the authors expand their discussion on this possibility?

2. Mouse monoclonal antibody 2A10 or human monoclonals CIS43 or mAb 10 were injected into mice prior to the assessment of sporozoite motility in the skin of these animals. Do the authors know if the concentration of the antibodies at the time of assay was similar in all mice? Which mice were used? Is it possible that the human monoclonal antibodies are cleared from the mouse at a faster rate than the mouse monoclonal antibody? Could this be why the human antibody was not as effective? Titers at time of assay or shortly thereafter would clarify this. Were C57BL/6 mice used? If yes, then could one use NSG mice instead as they could be less likely to remove the human monoclonal antibody?

3. Can the authors include a description of their sporozoite purification method?

Dear Zeljko,

Thank-you for all of your help with the final stretch of this process. I have now uploaded all the requested and changed files. The manuscript file is now in tracking mode but let me know if you want a cleaned up file as well. Below are all of your requests and our response to them. Please do not hesitate to contact me with any questions.

All the best,
Photini

1) Please address all the points raised by the referee #2

Referee #2 (Remarks for Author):

The authors have made their best effort to address the issues raised by this reviewer and also by the further reviewers. There are two minor comments I would like addressed.

1. Figure 5. The large decrease in sporozoites at the later timepoints for Pb is interesting. Pb is known to be a less host specific parasite than Py and Pf. CD81 for example is a hepatocyte receptor required by Py and Pf but not for Pb, and indeed Pb will infect non hepatocyte cell lines *in vitro*. Is it possible that the more rapid decrease in motility in the skin over time observed in Pb may be related to an early switch to inappropriate invasion, which is not observed in Py or Pf? Could the authors expand their discussion on this possibility?

The following text (in red) was added where this was discussed on page 12:

These data suggest that *P. yoelii* sporozoites have a longer period of time during which they are infectious post-inoculation. *These data suggest that P. yoelii sporozoites have a longer period of time during which they are infectious post-inoculation. In contrast, P. berghei sporozoite motility and blood vessel entry decrease significantly at later timepoints. This could be explained by a more rapid switch to a hepatocyte-invasion mode in P. berghei, a possibility that is consistent with its more promiscuous infectivity in vitro (Silvie et al, 2007) and could lead to a portion of the P. berghei inoculum prematurely switching to an invasion mode in the skin.*

2. Mouse monoclonal antibody 2A10 or human monoclonals CIS43 or mAb 10 were injected into mice prior to the assessment of sporozoite motility in the skin of these animals. Do the authors know if the concentration of the antibodies at the time of assay was similar in all mice? Which mice were used? Is it possible that the human monoclonal antibodies are cleared from the mouse at a faster rate than the mouse monoclonal antibody? Could this be why the human antibody was not as effective? Titers at time of assay or shortly thereafter would clarify this. Were C57BL/6 mice used? If yes, then could one use NSG mice instead as they could be less likely to remove the human monoclonal antibody?

The following text (in red) was added where this was discussed on page 10:

When tested in our intravital imaging assay with *P. falciparum* sporozoites, we found that both mAb 10 and CIS43 decreased the number of motile sporozoites and decreased blood vessel entry, however, not to the same degree as mAb 2A10 and not reaching statistical significance (Figure 8C).

This could be due to the fact that they are human and not mouse mAbs. These human mAbs could be cleared more rapidly from the blood circulation in the mouse, leading to lower antibody concentrations in the skin, a possibility that deserves further investigation.

3. Can the authors include a description of their sporozoite purification method?

The following statement was added to the relevant portion of the Methods on page 14:

No purification or centrifugation of sporozoites was performed as we found that these can significantly impact sporozoite viability.

The authors performed the requested editorial changes.

27th Jan 2021

Dear Dr. Sinnis,

We are pleased to inform you that your manuscript is accepted for publication.

Corresponding Author Name: Photini Sinnis

Manuscript Number: EMM-2019-11796-V2